# Liver gene expression and its rewiring in hepatic steatosis are controlled by PI3Kα-dependent hepatocyte signaling

Marion Régnier[1,2], Arnaud Polizzi[1], Tiffany Fougeray[1], Anne Fougerat[1], Prunelle Perrier[1], Karen Anderson[3], Yannick Lippi[1], Sarra Smati[1], Céline Lukowicz[1], Frédéric Lasserre[1], Edwin Fouche[1], Marine Huillet[1], Clémence Rives[1], Blandine Tramunt[4,5], Claire Naylies[1], Géraldine Garcia[1], Elodie Rousseau-Bacquié[1], Justine Bertrand-Michel[4,6], Cécile Canlet[1], Sylvie Chevolleau-Mege[1], Laurent Debrauwer[1], Christophe Heymes[4], Rémy Burcelin[4], Thierry Levade[7,8], Pierre Gourdy[4,5], Walter Wahli[1,9,10], Yuna Blum[11], Laurence Gamet-Payrastre[1], Sandrine Ellero-Simatos[1], Julie Guillermet-Guibert[7], Phillip Hawkins[3], Len Stephens[3], Catherine Postic[2], Alexandra Montagner[4], Nicolas Loiseau[1]*, Hervé Guillou[1]*

1 Toxalim (Research Center in Food Toxicology), INRAE, ENVT, INP-PURPAN, UMR1331, Université de Toulouse, Toulouse, France, 2 Université Paris Cité, Institut Cochin, CNRS, INSERM, Paris, France, 3 The Signaling Programme, The Babraham Institute, Cambridge, United Kingdom, 4 Institut des Maladies Métaboliques et Cardiovasculaires, I2MC, Université de Toulouse, Inserm, Toulouse, France, 5 Diabetology Department, CHU de Toulouse, Toulouse, France, 6 Metatoul-Lipidomic Facility, MetaboHUB, Institut des Maladies Métaboliques et Cardiovasculaires, I2MC, Université de Toulouse, Inserm, Toulouse, France, 7 Centre de Recherches en Cancérologie de Toulouse (CRCT), Inserm U1037, CNRS U5071, Université de Toulouse, Toulouse, France, 8 Laboratoire de Biochimie, CHU de Toulouse, Toulouse, France, 9 Lee Kong Chian School of Medicine, Nanyang Technological University Singapore, Singapore, Singapore, 10 Center for Integrative Genomics, Université de Lausanne, Lausanne, Switzerland, 11 Univ Rennes, CNRS, INSERM, IGDR (Institut de Génétique et Développement de Rennes) – UMR6290, ERL U1305, Rennes, France

* nicolas.loiseau@inrae.fr (NL); herve.guillou@inrae.fr (HG)

## Abstract

Insulin and other growth factors are key regulators of liver gene expression, including in metabolic diseases. Most of the phosphoinositide 3-kinase (PI3K) activity induced by insulin is considered to be dependent on PI3Kα. We used mice lacking p110α, the catalytic subunit of PI3Kα, to investigate its role in the regulation of liver gene expression in health and in metabolic dysfunction-associated steatotic liver disease (MASLD). The absence of hepatocyte PI3Kα reduced maximal insulin-induced PI3K activity and signaling, promoted glucose intolerance in lean mice and significantly regulated liver gene expression, including insulin-sensitive genes, in *ad libitum* feeding. Some of the defective regulation of gene expression in response to hepatocyte-restricted insulin receptor deletion was related to PI3Kα signaling. In addition, though PI3Kα deletion in hepatocytes promoted insulin resistance, it was protective against steatotic liver disease in diet-induced obesity. In the absence of hepatocyte PI3Kα, the effect of diet-induced obesity on liver gene expression was significantly altered, with changes in rhythmic gene expression in liver. Altogether,

**Data availability statement:** Microarray data and all experimental details are available in the Gene Expression Omnibus (GEO) database (GSE249340, GSE249341, GSE249343).

**Funding:** M.R. was supported by a PhD grant from Toulouse University. T.F. was supported by a PhD grant from Toulouse University and Institut National du Cancer (INCA). A.F. was supported by AgreenSkills and Région Occitanie. This work was supported by ANR (ANR-20-CE14-0035) Hepatomorphic (P.G., C.P., A.M., H.G.), by Région Occitanie (P.G., A.M., N.L., and H.G.), by Fondation de France (to H.G.) and by Fondation pour la Recherche Médicale (Equipe FRM EQU202303016327). The funders had no role in study design, data collection and analysis, decision to publish, or preparation of the manuscript.

**Competing interests:** The authors have declared that no competing interests exist.

**Abbreviations:** ALT, alanine transaminase; AST, aspartate transaminase; FFAs, free fatty acids; FDR, false discovery rate; GEO, gene expression omnibus; GSEA, gene set enrichment analysis; HFD, high fat diet; IR, insulin receptor; IRS, IR substrate; KD, ketogenic diet; MASLD, metabolic dysfunction-associated steatotic liver disease; qPCR, quantitative polymerase chain reaction; Shc, Src homology and collagen; TBP, TATA-box binding protein; PDK, phosphoinositide-dependent kinase; PH, pleckstrin homology; PI3K, phosphoinositide 3-kinase; ZT, zeitgeber time.

this study highlights the specific role of p110α in the control of liver gene expression in physiology and in the metabolic rewiring that occurs during MASLD.

## Introduction

The regulation of gene expression in response to growth factors and hormones is essential for the control of cell growth, survival, and metabolism. Activation of class I phosphonisotide-3 kinases (PI3Ks) is a critical signal transduction pathway used by cell-surface receptors to regulate intracellular events [1]. Receptors that access these signaling pathways include those that recognize growth factors and hormones such as insulin, a major regulator of metabolic homeostasis [2].

The concentration of insulin increases in response to a meal and controls the balance between fuel utilization and storage, maintaining glucose levels during fasting and feeding [3,4]. This occurs, in part, through the ability of insulin to regulate gene expression in a tissue- and cell-specific manner. There are three ways insulin can regulate gene expression via insulin receptors (IRs) [5]. Insulin binds to IRs present on the cell surface, activating their tyrosine kinase activity. This initiates a broad network of protein recruitment and phosphorylation, including the IR substrate (IRS) family, leading to activation of the PI3K pathway, as well as the Src homology and collagen (Shc) family and further activation of the Ras/MAP kinase pathway [5]. Stimulation of these pathways finally regulates transcription factors that control gene expression. Moreover, insulin-bound receptor can translocate to the nucleus to regulate gene expression [6].

Metabolic effects of insulin are generally considered to depend on the pathway involving PI3K. Class I PI3Ks synthesize the phospholipid PtdIns(3,4,5)P3 (PIP3). The increase in PIP3 concentration at the plasma membrane in response to PI3K activity triggers the recruitment of signaling molecules containing PIP3-specific Pleckstrin homology (PH) domains [1,7]. Phosphoinositide-dependent kinase (PDK) and the serine/threonine protein kinase AKT (also known as protein kinase B, PKB) possess PH domains able to bind PIP3. After the insulin-stimulated increase in PI3K activity, PDK and AKT translocate to the plasma membrane, leading to a local increase in their relative abundance [8,9]. At the plasma membrane, PDK phosphorylates AKT at Thr308. Additional Ser473 AKT phosphorylation dependent on mTORC2 occurs in a PI3K-dependent manner [8,10]. In the liver, insulin-induced AKT phosphorylation initiates a large number of downstream cellular responses, including glycogen synthesis, glycolysis, and lipogenesis, as well as inhibition of gluconeogenesis [4,11–14].

PI3Ks are heterodimers composed of a regulatory subunit tightly bound to a catalytic lipid kinase subunit. PI3Ks are named based on their catalytic subunit. Among the four different types of class I PI3Ks (α, β, γ, and δ), PI3Kα and PI3Kβ are expressed in hepatocytes. They share the same regulatory subunits but their activities depend on distinct catalytic subunits [1]. Previous studies have shown that most PI3K activity induced by insulin is dependent on PI3Kα. Deletion of the PI3Kα catalytic subunit (p110α) in mouse hepatocytes reduces PIP3 generation

and AKT activation upon insulin stimulation [15]. Mice lacking hepatocyte p110α show reduced insulin sensitivity and glucose tolerance, increased gluconeogenesis, and hypolipidemia that cannot be rescued by overexpression of the PI3Kβ catalytic subunit (p110β). Another report provided evidence that ablation of p110α, but not p110β, prevents hepatic steatosis associated with high fat diet (HFD)-induced obesity [16]. This protection from diet-induced steatosis is associated with decreased lipogenesis and reduced fatty acid uptake in hepatocytes. In contrast, other reports have shown that loss of p110α in hepatocytes is compensated by redundant PI3K activities dependent on p110β [17]. Moreover, low doses of a PI3Kα-selective inhibitor can successfully improve PI3Kα-related overgrowth syndrome without major metabolic effects [18,19].

In the present study, we analyzed the effect of hepatocyte p110α deletion on metabolic homeostasis in mice. Feeding is known to trigger changes in gene expression that depend both on IR-dependent and -independent processes in the liver [20,21], and physiological levels of insulin regulate a broad network of transcripts [22]. We performed an unbiased analysis to investigate the role of PI3Kα in the control of liver genome expression, including insulin-sensitive pathways in response to feeding. Obesity and type 2 diabetes are associated with metabolic dysfunction-associated steatotic liver disease (MASLD) [23], which is associated with extensive changes in hepatic signaling [24], genome expression [25–27], and metabolic pathways [27–29]. These changes include modification of the circadian liver rhythm, which occurs independently from changes in core clock gene expression [25,26,30,31]. Since the mechanisms underlying these changes are not elucidated, we also investigated the contribution of PI3Kα to the rewiring of liver gene expression and its rhythmicity that occurs in diet-induced obesity and MASLD. Taken together, our findings show that signaling dependent on PI3Kα is critical for the regulation of liver gene expression in both health and metabolic disease.

## Materials and methods

### Mice

*In vivo* studies were conducted under the European Union guidelines for the use and care of laboratory animals and were approved by an independent local ethics committee: CEEA-86: Comité d'éthique de Pharmacologie-Toxicologie de Toulouse-Midi Pyrénées en matière d'expérimentation animale (Toxcométhique). The specific approval number issued by the animal ethics committee to conduct the study is APAFIS#17430-2018110611093660 v3.

p110α$^{hep-/-}$ animals were created by mating floxed-p110α mice with C57BL/6J albumin-Cre transgenic mice to obtain albumin-Cre$^{+/-}$ p110α$^{flox/flox}$ mice (i.e., p110α$^{hep-/-}$ mice). The floxed-p110α mice were described previously [32]. In this mouse model, the deletion of p110α was shown to have no effect on the expression of other key components of the PI3K pathway such as p85 and p110β. The p110α deletion was confirmed by PCR and HotStar Taq DNA Polymerase (5 U/µl, Qiagen) using two primer pairs: FE1, 5′-GGATGCGGTCTTTATTGTC-3′ and FE4, 5′-TGGCATGCTGCCGAATTG-3′; ma9, ACACACTGCATCAATGGC and a5, GCTGCCGAATTGCTAGGTAAGC. The amplification step for the floxed mice were as follows: 94 °C for 3 min; followed by 20 cycles of touch down PCR comprising 94 °C for 1 min, 65 to 55 °C for 1 min 30 sec, and 72 °C for 1 min 30 sec; followed by 20 cycles of PCR comprising 94 °C for 1 min, 55 °C for 1 min 30 sec, and 72 °C for 1 min 30 sec; and a final cycle of 72 °C for 10 min. This reaction produced 634-bp, 544-bp, and 716-bp fragments, which represented the wild-type allele, the p110α sequence with an exon 18−19 deletion, and the floxed allele, respectively. The albumin-Cre allele was detected by PCR and HotStar Taq DNA Polymerase (5 U/µl, Qiagen) using the following primers: CreU, 5′-AGGTGTAGAGAAGGCACTTAG-3′ and CreD, 5′-CTAATCGCCATCTTCCAGCAGG-3′; G2lox7F, 5′-CCAATCCC TTGGTTCATGGTTGC-3′ and G2lox7R, 5′-CGTAAGGCCCAAGGAAGTCCTGC-3′). Amplification conditions to confirm the presence of CRE were as follows: 95 °C for 15 min; followed by 35 cycles of 94 °C for 1 min, 65 °C for 1 min, and 72 °C for 1 min; and a final cycle of 72 °C for 10 min. This reaction produced a 450-pb fragment representing the albumin-Cre allele.

Hepatocyte-specific IR knockout mice (IR$^{hep-/-}$) were generated by crossing animals carrying LoxP sites flanking the fourth exon of the IR gene (IR$^{lox/lox}$ stock number 006955; Jackson Laboratory, Bar Harbor, ME, USA) with C57BL/6J mice, which specifically express CRE recombinase in the liver under the control of the transthyretin promoter (TTR-CreTam mice), as described previously [20].

Mice were housed under controlled temperature (23 °C) and light (12-h light/12-h dark) conditions. All mice were males with free access to food (A04 U8220G1OR, Safe) and water. Albumin-Cre$^{-/-}$ floxed-p110α (p110α$^{hep+/+}$) and TTR-Cre$^{-/-}$ floxed IR (IR$^{hep+/+}$) littermates were used as controls. The mice that are used for the experiments described throughout this paper were usually 8–12 week-old, unless stated otherwise.

### *In vivo* experiments

**High fat diet.** Twelve-week-old p110α$^{hep+/+}$ and p110α$^{hep-/-}$ mice were fed a HFD with 60% calories from fat (D12492, Research Diet) or a control (CTRL) diet (D12450J, Research Diet) for 12 weeks. All mice had free access to food and water. At the end of the experiment, mice were fasted at Zeitgeber Time (ZT) 16 (with ZT0 being when the light is turned on and ZT12 when light is turned off) for 24 h or fed *ad libitum* prior to sacrifice. Mice were sacrificed at ZT16 ($n$ = 6–8 mice per group).

**Choline-deficient high fat diet.** Eight-week-old p110α$^{hep+/+}$ and p110α$^{hep-/-}$ mice were fed a choline-deficient HFD (CD-HFD) with 60% calories from fat (D05010403, Research Diet), a HFD with normal choline (A06071306, Research Diet), or a control (CTRL) diet (10% calories from fat) with normal choline (A08051501, Research Diet) for 11 weeks. Mice were sacrificed at ZT16 ($n$ = 6–8 mice per group).

**Ketogenic diet.** Twelve-week-old p110α$^{hep+/+}$ and p110α$^{hep-/-}$ mice were fed a CTRL diet (SAFE A04, Augy, France) containing 72.4% calories from carbohydrates, 8.3% from lipids, and 19.3% from proteins or a ketogenic diet (KD; TD.96355, Envigo, United States) containing 90.5% calories from lipids, 9.2% from proteins, and 0.3% from carbohydrates for 9 days. Mice were sacrificed at ZT16 ($n$ = 6–8 mice per group).

**Fasting/high glucose.** Sixteen-week-old p110α$^{hep+/+}$ and p110α$^{hep-/-}$ mice were fed a chow diet (SAFE A04, Augy, France). Half of the mice were given access to drinking water supplemented with 20% glucose (D-glucose, G8270, Sigma–Aldrich) 24 h prior to sacrifice. The other half of the mice were fasted 12 h prior to sacrifice. Mice were sacrificed at ZT16. Eight mice of each genotype were used for each experimental condition.

**Two-bottle preference assay and sucrose preference test.** Mice were acclimated to cages with two bottles of water 1 week before the challenge. Mice were then given access to bottles with either water or water containing 10% sucrose (S9378, Sigma–Aldrich) and consumption measured daily for 4 days. The positions of the bottles in the cages were switched every day during the challenge. Mice were fed a chow diet (SAFE A04, Augy, France) throughout the experiment. At the end of the challenge, blood was taken from fed mice to measure the plasma FGF21 level in response to the sucrose load. Sucrose preference was determined by the difference between water intake and sucrose-containing water intake.

**Fasting and fasting-refeeding.** Twelve-week-old p110α$^{hep+/+}$ and p110α$^{hep-/-}$ mice were divided into three groups. One group was fed *ad libitum* until sacrifice, another group was fasted for 24 h from ZT16, and the third group was fasted for 24 h and then was refed for 4 h with the addition of 200 g/L glucose in water ($n$ = 6 mice/genotype/experimental condition). All mice were sacrificed at ZT16.

**Circadian experiment.** Twelve-week-old p110α$^{hep+/+}$ and p110α$^{hep-/-}$ mice were fed a control diet (D12450J, Research Diet) or an HFD (D12492, research diet) for 16 weeks. At the end of the experiment, mice were sacrificed at different time points: ZT0, ZT4, ZT8, ZT12, ZT16, and ZT20 ($n$ = 6/genotype/experimental condition). All mice had free access to food and water until sacrifice.

***In vivo* insulin signaling.** Animals were fasted for 12 h from ZT0. At ZT12, the mice were anesthetized with isoflurane and ketamine (100 mg/kg body weight), followed by injection with insulin at 5 U/kg body weight (Umuline rapide, Lilly

Laboratories) via the inferior vena cava. Five minutes after the injection, the liver, adipose tissue, and muscles were excised and snap-frozen in liquid nitrogen. Anesthetized fasted mice served as a control for this experiment.

**Metabolic and physiological assays**

**Glucose, insulin, and pyruvate tolerance tests.** For the glucose tolerance test, mice were fasted for 6 h and received an oral (2 g/kg body weight) glucose load (G7021, Sigma–Aldrich). For the insulin tolerance test, mice were fasted for 6 h and received insulin (Umuline rapide, Lilly Laboratories) by intraperitoneal injection (0.2 U/kg). For the pyruvate tolerance test, mice were fasted for 24 h and received pyruvate by intraperitoneal injection (20% in NaCl; 2 g/kg). For all tolerance tests, blood glucose was measured at the tail vein using an AccuCheck Performa glucometer (Roche Diagnostics) at T-15 (15 min before injection), T0, and 15, 30, 45, 60, 90, and 120 min after injection (and 150 min for pyruvate tolerance test).

**Circulating glucose, ketone bodies, and other biochemical analysis.** Blood glucose was measured using an Accu-Chek Go glucometer (Roche Diagnostics). The β-hydroxybutyrate content was measured at different fasting time points using Optium β-ketone test strips carrying Optium Xceed sensors (Abbott Diabetes Care). Aspartate transaminase (AST), alanine transaminase (ALT), free fatty acids (FFAs), triglycerides, total cholesterol, LDL cholesterol, and HDL cholesterol levels were determined from plasma samples using a COBASMIRA° biochemical analyzer (Anexplo facility).

**ELISA.** Plasma FGF21 and insulin were assayed using the rat/mouse FGF21 ELISA kit (EMD Millipore) and the ultrasensitive mouse insulin ELISA kit (Crystal Chem), respectively, following the manufacturers' instructions.

**Blood and tissue sampling.** Prior to sacrifice, the submandibular vein was lanced and blood collected in EDTA-coated tubes (BD Microtainer, K2E tubes). Plasma was collected by centrifugation (1,500× $g$, 10 min, 4 °C) and stored at −80 °C. Following sacrifice by cervical dissociation, organs were removed, weighed, dissected, and used for histological analyses or snap-frozen in liquid nitrogen and stored at −80 °C.

**Analysis of insulin signaling in animal tissue by western blot.** Western blot analyses were conducted under different conditions. In the first experiment, mice were fasted for 12 h from ZT0 and received vena cava injections of insulin (Umulin rapide, Lilly Laboratories) at ZT12. Mice were sacrificed 5 min after insulin injection. In a second experiment, mice were fasted for 24 h and refed for 4 h with 200 g/L of glucose in water. Fasted mice were used as a control for each experiment. Tissues were immediately removed and homogenized in RIPA buffer by sonication (50 mM Tris HCl, pH 7.4, 150 mM NaCl, 2 mM EDTA, 0.1% SDS, 1 mM PMSF, 1% NP40, 0.25% sodium deoxycholate, proteinase, and phosphatase inhibitors). Particulates were removed by centrifugation at 13,000 $g$ for 30 min. The protein concentration in each lysate was measured using a BC Assay Protein Quantitation Kit (Interchim).

Proteins were separated by SDS-polyacrylamide gel electrophoresis and transferred to a nitrocellulose membrane. Immunodetection was performed using the relevant primary antibody overnight at 4 °C: anti-pSer473Akt (4058), anti-pThr308Akt (2965), anti-total Akt (9272), anti-pThr389p70 S6 kinase (9205), anti-p110α (4255), anti-pSer9GSK-3β (9336), or anti-β-actin (all Cell Signaling, 1:1000 dilution). Signals were acquired using Chemidoc (Bio-Rad) and quantified by ImageJ software. Signal intensities obtained with phospho-specific antibodies were normalized to those obtained with antibodies against total proteins. Signal obtained with p110α-specific antibodies were normalized to β-actin.

**PIP3 quantification.** Following the vena cava injection of insulin, the liver was dissected and snap-frozen in liquid nitrogen. Mass spectrometry was used to measure inositol lipid levels as described previously [33] using a QTRAP 4000 (AB Sciex) mass spectrometer and employing the lipid extraction and derivatization method.

**Liver glycogen quantification.** For liver glycogen content measurement, frozen liver tissues were weighed and homogenized into 4% (v/v) ice-cold perchloric acid (50 mg/mL). After centrifugation for 10 min at 8,000 rpm and 4 °C, the acid supernatant was neutralized, and glycogen from the neutralized supernatant was digested for 1 h at 55 °C without or with α-(1-4)-(1-6)-amyloglucosidase (10 mg/mL, Roche Applied Science) to hydrolyze glycogen. The amount of supernatant required for hydrolysis was dependent on nutritional status (100 μL for fasted mice and 40 μL of supernatant

for fed or refed mice). Following hydrolysis, the glucose released was quantified by measuring at 25 °C the appearance of NADPH at 340 nm: 250 μL final volume containing 100 mM Tris (pH 8.5), 15 mM MgCl2, 0.5 mM NADP, and 0.7 mM ATP was added, followed by the addition of 1 unit of glucose-6-phosphate dehydrogenase (Roche Diagnostics) for 10 min. After a first lecture at 340 nm, 1.5 units of hexokinase (Roche Diagnostics) were added for 15 min before measuring the final glucose released at 340 nm. Results were expressed as μmoles of glycogen per μg of proteins per milligram of liver.

**Gene expression.** Total cellular RNA was extracted using Tri reagent (Molecular Research Center). Total RNA samples (2 μg) were reverse-transcribed using the High-capacity cDNA Reverse Transcription Kit (Applied Biosystems) for real-time quantitative polymerase chain reaction (qPCR) analyses. The primers for Sybr Green assays are presented in S1 Table. Amplification was performed on a Stratagene Mx3005P (Agilent Technology). The qPCR data were normalized to the level of TATA-box binding protein (TBP) mRNA and analyzed by LinRegPCR.

**Microarray.** Gene expression analysis was performed as previously described [20]. Transcriptome profiling was performed at the GeT-TRIX facility (Génotoul, Génopole, Toulouse, Occitanie) using Agilent SurePrint G3 Mouse GE 8x60K (Design 074809) according to the manufacturer's instructions [34–36]. For each sample, cyanine-3 (Cy3) labeled cRNA was prepared from 200 ng of total RNA using the One-Color Quick Amp Labeling kit (Agilent) according to the manufacturer's instructions, followed by Agencourt RNAClean XP (Agencourt Bioscience Corporation, Beverly, MA). Dye incorporation and cRNA yield were checked using Dopsense 96 UV/VIS droplet reader (Trinean, Belgium). A total of 600 ng of the Cy3-labeled cRNA was hybridized on the microarray slides following the manufacturer's instructions. Immediately after washing, the slides were scanned on an Agilent G2505C Microarray Scanner using Agilent Scan Control A.8.5.1 software and the fluorescence signal extracted using Agilent Feature Extraction software v10.10.1.1 with default parameters. Microarray data and all experimental details are available in the Gene Expression Omnibus (GEO) database (GSE249340, GSE249341, GSE249343).

**Histology.** Paraformaldehyde-fixed, paraffin-embedded liver tissue was sliced into 3-μm-thick sections and stained with hematoxylin and eosin for histopathological analysis. The staining was visualized under a Leica DM4000 B microscope equipped with a Leica DFC450 C camera.

[1]**H-NMR based metabolomics.** Metabolomic profiling was performed as described previously [37]. All [1]H-NMR spectra were obtained on a Bruker DRX-600-Avance NMR spectrometer (Bruker, Wissembourg, France) using the AXIOM metabolomics platform (MetaToul) operating at 600. 13 MHz for the [1]H resonance frequency and an inverse detection 5-mm [1]H-[13]C-[15]N cryoprobe attached to a cryoplatform (the preamplifier cooling unit).

The [1]H-NMR spectra were acquired at 300 K using a standard one-dimensional noesypr1D pulse sequence with water presatration and a total spin-echo delay (2 nτ) of 100 ms. A total of 128 transients were collected into 64,000 data points using a spectral width of 12 ppm, relaxation delay of 2.5 s, and acquisition time of 2.28 s. The [1]H-[1]H COSY, [1]H-[1]H TOCSY, and [1]H-[13]C HSQC were obtained for each biological matrix in one representative sample for metabolite identification.

Data were analyzed by applying an exponential window function with 0.3-Hz line broadening prior to Fourier transformation. The resulting spectra were phased, baseline corrected, and calibrated to TSP (δ 0.00 ppm) manually using Mnova NMR (v9.0, Mestrelab Research). The spectra were subsequently imported into MatLab (R2014a, MathsWorks, Inc.). All data were analyzed using full-resolution spectra. The region containing water resonance (δ 4.6–5.2 ppm) was removed, and the spectra were normalized to the probabilistic quotient [38] and aligned using a previously published function [39].

Data were mean-centered and scaled using the unit variance scaling prior to analysis with orthogonal projection on latent structure-discriminant analysis (O-PLS-DA). The [1]H-NMR data were used as independent variables (X matrix) and regressed against a dummy matrix (Y matrix) indicating the class of samples. The O-PLS-derived model was evaluated for goodness of prediction (Q2Y value) using n-fold cross-validation, where n depends on the sample size. To identify metabolites responsible for discrimination between the groups, the O-PLS-DA correlation coefficients ($r2$) were calculated for each variable and back-scaled into a spectral domain so that the shapes of the NMR spectra and signs of the coefficients were preserved [40,41]. The weights of the variables were color-coded according to the square of the O-PLS-DA

correlation coefficients. Correlation coefficients extracted from significant models were filtered so that only significant correlations above the threshold defined by Pearson's critical correlation coefficient ($P < 0.05$; $|r| > 0.7$; $n = 12$ per group) were considered significant.

**Reporter metabolite analysis.** Reporter metabolite analyses were performed to investigate the metabolites affected by transcriptional changes in the absence of p110α using the PIANO package, which enriches the gene set analysis of genome-wide data by incorporating the directionality of gene expression and combining statistical hypotheses and methods in conjunction with a previously established genome-scale metabolic model of the liver, iHepatocyte2322 [42].

**Liver neutral lipids analysis.** Tissue samples were homogenized in methanol/5 mM EGTA (2:1, v/v) and lipids (equivalent to 2 mg of tissue) extracted according to the Bligh–Dyer method [43] with chloroform/methanol/water (2.5:2.5:2 v/v/v) in the presence of the following internal standards: glyceryl trinonadecanoate, stigmasterol, and cholesteryl heptadecanoate (Sigma). Triglycerides, free cholesterol, and cholesterol esters were analyzed by gas-liquid chromatography on a Focus Thermo Electron system equipped with a Zebron-1 Phenomenex fused-silica capillary column (5 m, 0.25 mm i.e., 0.25 mm film thickness). The oven temperature was programmed to increase from 200 to 350 °C at 5 °Cmin⁻¹, and the carrier gas was hydrogen (0.5 bar). Injector and detector temperatures were 315 °C and 345 °C, respectively. All of the quantitative calculations were based on the peak area ratios relative to the internal standards [44].

**Liver fatty acid analysis.** Tissue samples were weighed and homogenized using a Fastprep system (40 s) in methanol/5 mM EGTA (2:1, v/v). An aliquot corresponding to 1 mg of tissue was sampled and 2 µg TG17 (glyceryl triheptadecanoate, Sigma–Aldrich, les Ulis, France) added as internal standard to verify the completeness of hydrolysis. After hydrolysis in KOH-methanol (1.5 M) for 30 min at 56 °C, FAs were transmethylated with 1 mL BF3-methanol, 10% wt. (60 min at 80 °C). Once cooled down, 1 mL Milli-Q water and 2 mL heptane was added to the methylated FAs and the mixture manually shaken. After centrifugation (500$g$, 5 min), the upper layer containing FA methyl esters (FAMEs) was transferred to a glass tube and evaporated to dryness. Heptane (200 µL) was then added and the sample transferred to a vial. FAMEs were analyzed on a TRACE 1,310 gas chromatograph (Thermo Scientific, Les Ulis, France) equipped with a split-splitless injector operated in the splitless mode and a flame-ionization detector. FAMEs were separated on a FAME-WAX column (30 m, 0.32 mm internal diameter, 0.25 µm film thickness) from Restek (Lisses, France) using helium as carrier gas at a constant flow rate of 1.0 mL.min⁻¹. The injector temperature was set at 225 °C and the oven temperature was programmed as follows: 1 min isothermal step at 130 °C, from 130 to 245 °C at 2 °C.min⁻¹ and then 8 min at 245 °C. FAMEs were identified by comparing sample retention times to those of commercial standard mixtures (Menhaden oil and Food Industry FAME Mix, Restek) using Xcalibur 2.2 software (Thermo Scientific).

**Liver phospholipid and sphingolipid analysis.** *Chemicals and reagents:* The liquid chromatography solvent, acetonitrile, was HPLC-grade and purchased from Acros Organics. Ammonium formate (>99%) was supplied by Sigma–Aldrich. Synthetic lipid standards (Cer d18:1/18:0, Cer d18:1/15:0, PE 12:0/12:0, PE 16:0/16:0, PC 13:0/13:0, PC 16:0/16:0, SM d18:1/18:0, SM d18:1/12:0) were purchased from Avanti Polar Lipids.

*Lipid extraction:* Lipids were extracted from the liver (1 mg) as described previously [45] using dichloromethane/methanol (2% acetic acid)/water (2.5:2.5:2 v/v/v). Internal standards were added (Cer d18:1/15:0, 16 ng; PE 12:0/12:0, 180 ng; PC 13:0/13:0, 16 ng; SM d18:1/12:0, 16 ng; PI 16:0/17:0, 30 ng; PS 12:0/12:0, 156.25 ng) and the solution centrifuged at 1500 rpm for 3 min. The organic phase was collected and dried under azote, then dissolved in 50 µl MeOH. Sample solutions were analyzed by an Agilent 1290 UPLC system coupled to a G6460 triple quadripole spectrometer (Agilent Technologies). MassHunter software was used for data acquisition and analysis. A Kinetex HILIC column (Phenomenex, 50 × 4.6 mm, 2.6 µm) was used for LC separation. The column temperature was maintained at 40 °C. The mobile phase, A, was acetonitrile and the B phase 10 mM ammonium formate in water at pH 3.2. The gradient was as follows: 10% to 30% B in 10 min; 100% B from 10 to 12 min; and then back to 10% B at 13 min for 1 min to re-equilibrate prior to the next injection. The flow rate of the mobile phase was 0.3 ml.min⁻¹, and the injection volume was 5 µl. An electrospray source was employed in positive (for Cer,

PE, PC, and SM analysis) or negative ion mode (for PI and PS analysis). The collision gas was nitrogen. The needle voltage was set at +4000 V. Several scan modes were used. First, to obtain the naturally different masses of different species, we analyzed cell lipid extracts with a precursor ion scan at 184 m/z, 241 m/z, and 264 m/z for PC/SM, PI, and Cer, respectively. We performed a neutral loss scan at 141 and 87 m/z for PE and PS, respectively. The collision energy optimums for Cer, PE, PC, SM, PI, and PS were 25 eV, 20 eV, 30 eV, 25 eV, 45 eV, and 22 eV, respectively. The corresponding SRM transitions were used to quantify different phospholipid species for each class. Two 9 MRM acquisitions were necessary due to important differences between phospholipid classes. We used QqQ Quantitative (vB.05.00) and Qualitative Analysis software (vB.04.00). All of the quantitative calculations were based on the peak area ratios relative to the internal standards [46].

### Statistical analysis

Biochemical, qPCR, and phenotypic data were analyzed using GraphPad software. Differential effects were assessed on $\log_2$ transformed data by ANOVA followed by Sidak post-hoc tests. *P*-values <0.05 were considered significant.

Hierarchical clustering of lipid quantification data was performed using R (R Development Core Team (2022) R Core Team (2022). R: A Language and Environment for Statistical Computing. R Foundation for Statistical Computing, Vienna, Austria, https://www.R-project.org/) with the pheatmap package [47]. Hierarchical clustering on $\log_2$ transformed data was applied to the samples and lipids using 1-Pearson correlation coefficient as distance and Ward's criterion (Ward.D2) for agglomeration. All of the data represented on the heat map had adjusted P-values <0.05 for one or more comparisons performed with an analysis of variance and is scaled by row.

Microarray data were analyzed using R and Bioconductor packages (www.bioconductor.org, v 3.0) as described in GEO accession (GSE249340, GSE249341, GSE249343). Raw data (median signal intensity) were filtered, $\log_2$ transformed, corrected for batch effects (microarray washing bath), and normalized using the quantile method.

A model was fitted using the limma lmFit function considering array weights using the arrayWeights function. Pair-wise comparisons between biological conditions were applied using specific contrasts. A correction for multiple testing was applied using the Benjamini–Hochberg procedure for the false discovery rate (FDR). Probes with FDR ≤ 0.05 were considered to be differentially expressed between conditions.

Hierarchical clustering was applied to the samples and the differentially expressed probes using 1-Pearson correlation coefficient as distance and Ward's criterion for agglomeration. The clustering results are illustrated as a heatmap of the expression signals. Gene set enrichment analysis (GSEA) was performed using the ViSEAGO R package [48] with signed $-\log_{10}$ adjusted *p*-values as the score and 10,000 permutations of random sets of 101 genes: $score = -\log_{10}(adj.p.value) * \sqrt{logFC^2} / logFC$. GO categories with a minimal size of 50, $P < 0.01$, and $\log_2$ err > 0.5 were considered significant. For GO enrichment, hypergeometric tests were performed over genes selected in GO categories extracted from the org.Mm.eg.db R package version 3.17.0. against all expressed genes. $P < 0.05$ was considered significant. Principal component analysis (PCA) was performed on unscaled $\log_2$ transformed data using the R package FactoMineR [49,50]. Circadian gene expression profiles during a 24 h period was determined with R package dryR [51].

## Results

### Hepatocyte p110α deficiency leads to a 50% decrease in PIP3 *in vivo* and altered glucose homeostasis

Hepatocyte-specific *p110α* knockout mice (*p110α^flox/flox^-albumin-Cre^+/−^* i.e., *p110α^hep−/−^*) were generated by crossing C57Bl/6J *p110α^flox/flox^* mice with *albumin-Cre* mice in the same genetic background (S1 Fig). Hepatocyte-specific *p110α* deletion was validated by PCR analysis of *p110α* floxed (*p110α^hep+/+^*) and Albumin-Cre (Albumin-Cre^+/−^) genes from *p110α^hep+/+^* and *p110α^hep−/−^* mice using DNA from different tissues (liver, white adipose tissue, brown adipose tissue, and tail; **Fig 1A**). This reaction produced 544-bp fragments specifically in the livers of *p110α^hep−/−^* mice and corresponded to the deleted sequence.

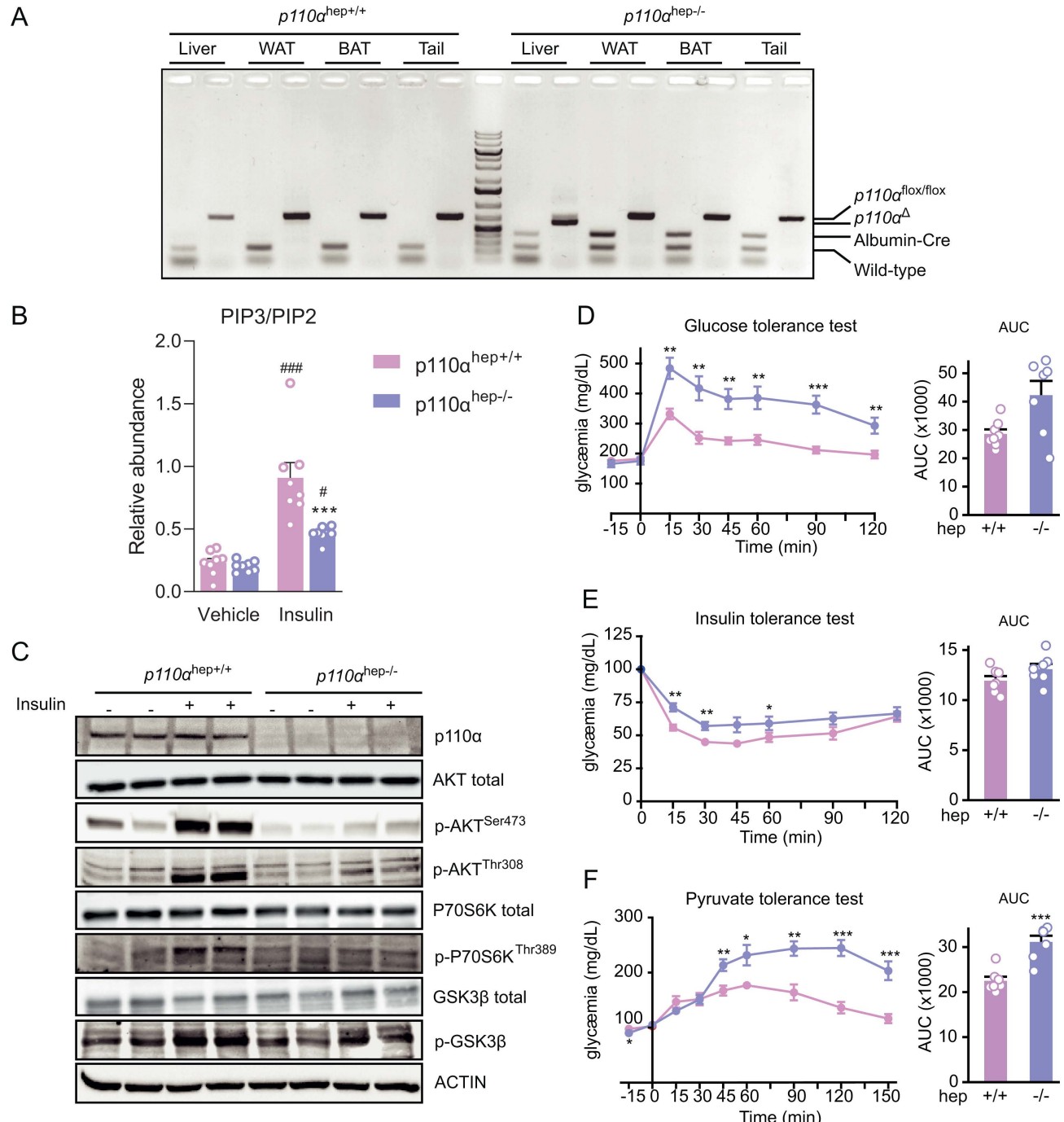

**Fig 1. Characterization of the hepatocyte-specific *p110α* knockout mouse model.** (A) PCR analysis of *p110α* floxed (*p110α^flox/flox*), *p110α* deleted (p110α^Δ), and Albumin-Cre (Albumin-Cre^+/−) genes in control (*p110α^hep+/+*) or liver knockout (*p110α^hep−/−*) mice using genomic DNA from the liver, white adipose tissue (WAT), brown adipose tissue (BAT), and tail. (B) Liver PIP3/PIP2 ratio determined by LC/MS for *p110α^hep+/+* and *p110α^hep−/−* mice that were fasted and then treated or not with insulin (5 U/kg) by vena cava injection (*n* = 8 mice/group). (C) Western blots of liver protein extracts evaluating the phosphorylation of AKT, p70S6K, and GSK3β in the mice in (B). **(D–F)** Glucose (D), insulin (E), and pyruvate **(F)** tolerance tests and corresponding AUC (*n* = 6 mice/genotype). Data information: In **(B, D–F)**, data are presented as mean ± SEM. ^##*P* ≤ 0.01 and ^###*P* ≤ 0.005 for treatment effect; **P* ≤ 0.01 and ****P* ≤ 0.005 for genotype effect. The numerical values underlying the panels for this figure can be found in S1 Data.

Loss of hepatocyte *p110α* resulted in decreased insulin signaling, which was associated with an altered PIP3/PIP2 ratio in hepatocytes *in vivo* (**Fig 1B**). While the loss of *p110α* has no effect on baseline PI3K activity, it reduces maximal insulin-stimulated PI3K activity. The *p110α$^{hep+/+}$* mice had a 4-fold increase in the PIP3/PIP2 ratio in hepatocytes 5 min after insulin stimulation, but this maximal induction was reduced by 50% in *p110α$^{hep-/-}$* mice. Importantly, upon stimulation with insulin, we could only detect PIP3 (38:4) while a broader variety of PIP2 molecular species were detected (S2A and S2B Fig).

In addition to a lack of p110α protein expression, *p110α$^{hep-/-}$* mice had a significant reduction in the phosphorylation of p-Akt $^{Ser473}$, p-Akt $^{Thr308}$, p-70S6 $^{Thr389}$ kinase and to a lesser extent p-GSK-3β $^{Ser9}$, 5 min after insulin stimulation (**Figs 1C**, S2C). These results support a critical role of hepatocyte p110α in part of the PIP3-mediated insulin signaling.

We also confirmed that the liver-specific absence of *p110α* leads to impaired glucose tolerance and reduced insulin sensitivity after glucose and insulin loading, respectively (**Fig 1D**,**1E**). The *p110α$^{hep-/-}$* mice also exhibited increased glucose production during the pyruvate tolerance test (**Fig 1F**). These results indicate that a 50% reduction in PIP3 production in *p110α$^{hep-/-}$* mice is associated with marked glucose intolerance. This phenotype may be due, at least in part, to defective inhibition of gluconeogenesis.

## Hepatocyte p110α signaling influences liver growth and liver gene expression in response to metabolic challenges

To investigate the role of hepatocyte p110α-dependent signaling in metabolic homeostasis, we explored the effects of feeding and fasting in *p110α$^{hep+/+}$* and *p110α$^{hep-/-}$* mice. First, we observed that *p110α$^{hep-/-}$* mice exhibited an altered response to feeding, characterized by elevated plasma insulin levels and a reduction in liver weight (**Fig 2A**). Plasma glucose was reduced in fasting mice while hepatic glycogen was increased in fed mice. The *p110α$^{hep+/+}$* and *p110α$^{hep-/-}$* mice showed no difference in plasma glucose. Glycogen levels were similar in both *p110α$^{hep+/+}$* and *p110α$^{hep-/-}$* mice upon feeding (**Fig 2A**). Next, gene expression profiling revealed that 46% of genes sensitive to feeding ($n$ = 762) are dependent on p110α (**Fig 2B**). Some of these genes ($n$ = 352) are differentially expressed between fed *p110α$^{hep+/+}$* and *p110α$^{hep-/-}$* mice (FC > 1.5 and $P$ < 0.05, **Figs 2C** and S3A). To further investigate the contribution of p110α to insulin signaling in fed mice, we used hepatic gene expression profiles obtained from C57BL/6J mice treated with physiological levels of insulin [22] and compared the data to the p110α-sensitive gene clusters (**Fig 2D**). We identified 33 hepatic genes that were up-regulated in the absence of p110α and are known to be repressed by insulin *in vivo*. Interestingly, we also identified 77 insulin-inducible genes that were down-regulated in fed mice lacking hepatocyte p110α.

In line with a significant role of hepatocyte p110α in the control of insulin-sensitive liver gene expression, bioinformatic analysis predicted SREBP1-c and ChREBP as the two main transcription factors altered in the absence of p110α (**Fig 2E**). In agreement, mRNA expression of *Pnpla3*, *Scd1*, and *Fsp27* was decreased in the livers of *p110α$^{hep-/-}$* mice when compared to their littermates (**Fig 2F**). In contrast, FOXO1 was the only transcription factor predicted to be related to increased gene expression in *p110α$^{hep-/-}$* mice. Among the genes for which we observed a significant increase in mRNA expression, we found hepatokines *Igfbp1*, *Igfbp2*, and *Enho*.

Finally, we performed unbiased hepatic metabolomics profiling of aqueous metabolites using $^1$H-NMR to investigate whether transcriptomic changes were coupled with metabolomic adjustments. Analysis of discriminant metabolites revealed higher betaine and lower lactate levels in the absence of *p110α* (S3B Fig). Transcriptomic and metabolomic data were then pooled in a consensus genome-scale metabolic model called iHepatocytes. The resulting metabolic network highlighted the metabolites that were predicted to be significantly different in the absence of hepatocyte *p110α* (**Fig 2G**). This analysis corroborated previous results and underscored the critical role of p110α-dependent signaling in glycolysis, pyruvate metabolism, and *de novo* fatty acid synthesis. In agreement with this analysis, we found that palmitate (C16:0) and oleate (C18:1n-9), two major products of *de novo* fatty acid biosynthesis, were reduced in the livers of *p110α$^{hep-/-}$* mice (**Fig 2H**).

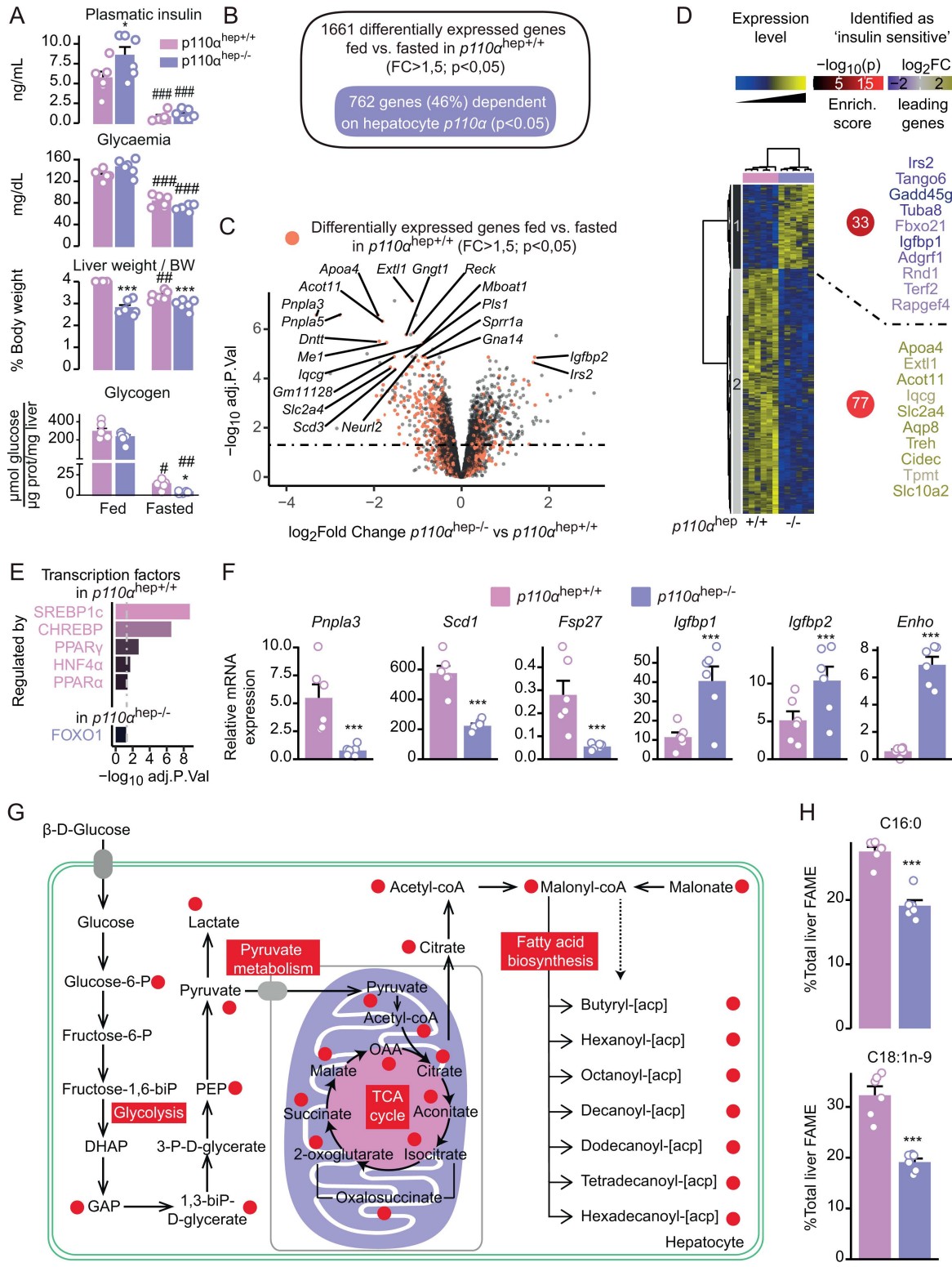

**Fig 2. Absence of p110α -dependent signaling in hepatocytes affects liver growth, glucose homeostasis, and the regulation of liver gene expression.** (A) Plasma insulin and glucose levels, relative liver weights and glycogen content in *p110α*^hep+/+^ and *p110α*^hep−/−^ mice that were fed or fasted for 24 h and processed at ZT16 (*n* = 6 mice/group). (B) Venn diagram representing the genes regulated in a p110α-dependent manner in response to

the nutritional status. (C) Volcano plot representing the differentially expressed genes in the liver of $p110\alpha^{hep-/-}$ versus $p110\alpha^{hep+/+}$ fed mice. Red dots represent hepatic genes that are differentially expressed between fasting and feeding in $p110\alpha^{hep+/+}$ mice. (D) Left: Heatmap illustrating microarray data for liver samples from fed $p110\alpha^{hep+/+}$ and $p110\alpha^{hep-/-}$ mice. Right: Enrichment of insulin-sensitive genes as determined by [22]. The color of the circle and the gene names are relative to the percentage of all identified insulin-sensitive genes within a cluster and the $P$-value of the hypergeometric test. (E) Top list of the transcription factors up-regulated (top) and down-regulated (bottom) by $p110\alpha$ hepatocyte deletion as determined by TRRUST. (F) Relative expression of *Pnpla3*, *Scd1*, *Fsp27*, *Igfbp1*, *Igfbp2*, and *Enho* measured by qPCR in the livers of $p110\alpha^{hep+/+}$ and $p110\alpha^{hep-/-}$ mice under fed conditions ($n$ = 6 mice/group). (G) Profile of p110α-related metabolites based on iHepatocyte genome scale modeling visualized on an integrative representation of glycolysis, pyruvate metabolism, and fatty acid biosynthesis. Red circles represent metabolites significantly altered ($P < 0.01$) in the absence of $p110\alpha$ in the fed state. (H) Relative hepatic abundance of fatty acid methyl ester (FAME) C16:0 and C18:1n-9 determined by GC/MS. Data information: Data are presented as mean ± SEM. ***$P \le 0.005$ for genotype effect. The numerical values underlying the panels for this figure can be found in S2 Data which includes the full gene expression analysis for this figure in the sheet "Microarray Fig 2D".

## Hepatocyte p110α is dispensable for carbohydrate and fatty acid sensing by hepatocytes

Bioinformatic analysis suggested that not only SREBP1 and FOXO1, but also ChREBP, PPARγ, PPARα, and HNF4α target genes may be sensitive to the expression of p110α in hepatocytes (**Fig 2E**). ChREBP is a key regulator of carbohydrate sensing, whereas the PPARs and HNF4α are involved in fatty acid sensing. Therefore, we questioned whether hepatocyte p110α may act not only as an insulin signaling intermediate, but also in carbohydrate- and fatty acid-mediated regulation of gene expression.

To address the possible role of p110α in the regulation of ChREBP-mediated gene expression, we performed an acute glucose challenge in which mice were either fasted or fed *ad libitum* with access to chow diet and 20% glucose solution in water for 24 h. Upon glucose challenge, $p110\alpha^{hep-/-}$ mice exhibited massive hyperglycemia (**Fig 3A**) and elevated ChREBP expression and activity, as revealed by the expression of *Chrebpα*, *Chrebpβ*, and *Lpk* in response to high glucose load (**Fig 3B**). In contrast, the glucose-mediated induction of SREBP1-c target genes, including *Gck*, *Acly,* and *Fasn,* was markedly reduced in the absence of hepatocyte p110α (**Figs 3B** and **S4A**). ChREBP is also a critical regulator of FGF21 hepatic expression, which controls sweet preference *in vivo*. We therefore investigated whether hepatocyte p110α affects sweet preference using the two-bottle preference assay. Absence of hepatocyte $p110\alpha$-dependent signaling did not alter sucrose preference or plasma glucose levels over the 4-day experimental period (**Fig 3C**). The circulating level of FGF21, which regulates sucrose intake under the control of Chrebp [52], was similarly increased in $p110\alpha^{hep+/+}$ and $p110\alpha^{hep-/-}$ mice (**Fig 3D**). Taken together, these findings indicate that p110α is not required for ChREBP-mediated regulation of gene expression in response to dietary carbohydrates and the control of sweet preference *in vivo*.

Next, we tested whether hepatocyte p110α-dependent signaling affects fatty acid sensing using two different challenges. First, we challenged mice with fasting and refeeding. In line with the role of p110α in insulin signaling, we observed that, upon refeeding, $p110\alpha^{hep-/-}$ mice exhibited high plasma levels of glucose and insulin, without any difference in liver glycogen content (**Fig 3E**). Metabolic differences induced by fasting and feeding were associated with heterogenous responses in the phosphorylation of insulin-sensitive AKT, S6K and GSK3 (**Figs 3F**, **S4B**). During fasting, changes in liver gene expression occur in response to adipose tissue lipolysis and regulate FGF21 production and ketogenesis through PPARα [53]. Therefore, we measured the mRNA expression of representative PPARα targets in the liver, including *Vnn1*, *Cyp4a14*, and *Fgf21*. Fasted $p110\alpha^{hep-/-}$ mice had increased mRNA expression of PPARα target genes, suggesting high fatty acid influx. Moreover, refeeding inhibited PPARα activity in a similar manner in both $p110\alpha^{hep+/+}$ and $p110\alpha^{hep-/-}$ mice, indicating that hepatocyte *p110α* is not required for the inhibition of PPARα activity during refeeding (**Figs 3G** and **S4C Fig**). In agreement, plasma acylcarnitine levels, which reflect fatty acid β-oxidation, were found similarly induced in response to fasting and inhibited by refeeding in both $p110\alpha^{hep+/+}$ and $p110\alpha^{hep-/-}$ mice (**Fig 3H**). We also tested the kinetics of glucose and ketone body levels in the transition between fasting and refeeding. Although $p110\alpha^{hep-/-}$ mice exhibited mild hyperglycemia 30 min after refeeding, ketone body levels remained similar to those of $p110\alpha^{hep+/+}$ mice, suggesting that refeeding rapidly inhibits liver fatty acid catabolism and ketone body production in both $p110\alpha^{hep+/+}$ and $p110\alpha^{hep-/-}$ mice (**Fig 3I**).

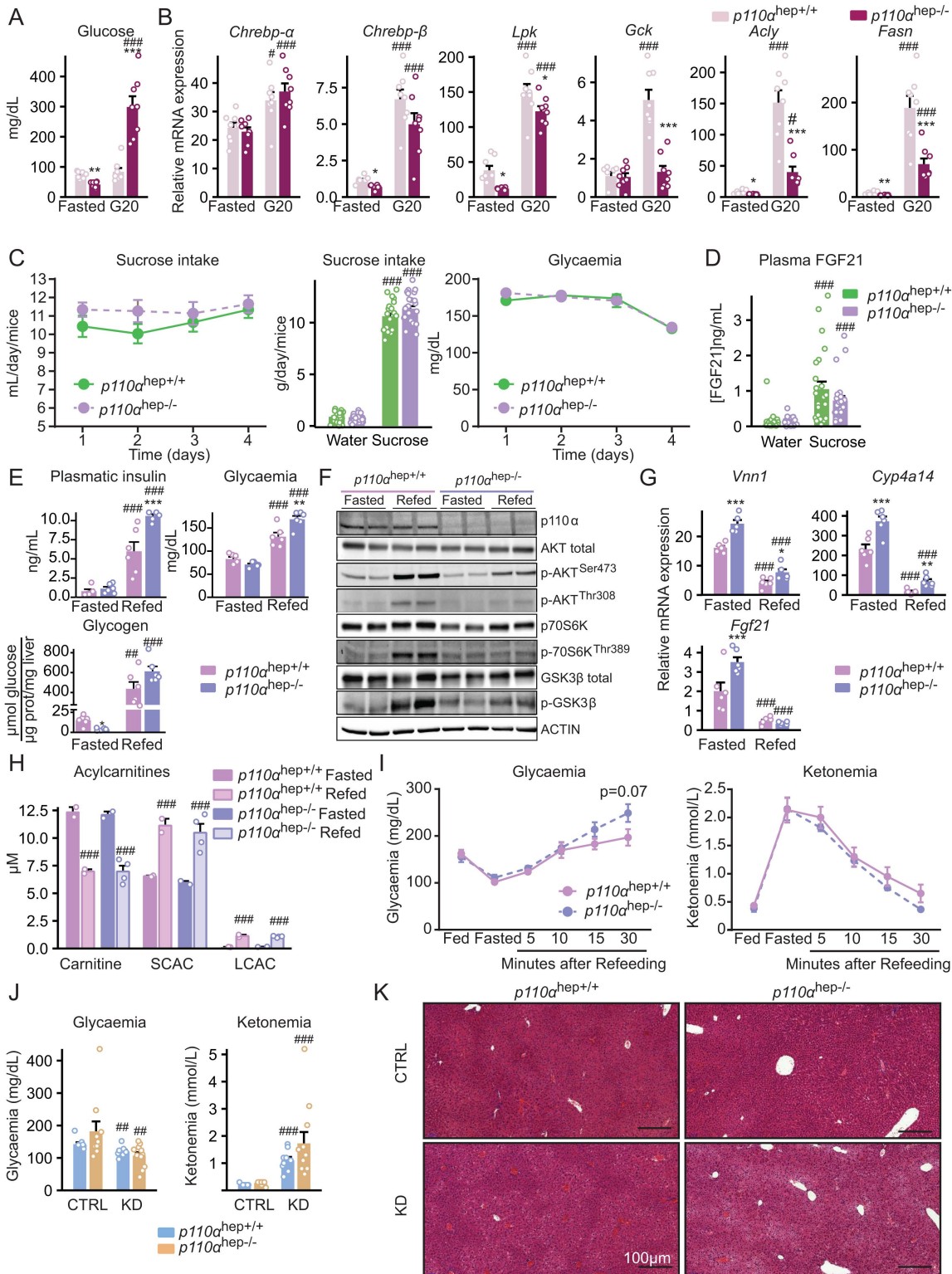

**Fig 3. Hepatocyte *p110α* is required for insulin signaling but not for glucose and fatty acid sensing. (A)** *p110α*<sup>hep+/+</sup> and *p110α*<sup>hep−/−</sup> mice were fasted or fed a chow diet supplemented with glucose (20%) in drinking water (*n* = 8 mice/group) and the plasma glucose levels measured. **(B)** Relative liver gene expression of *Chrebp-α*, *Chrebp-β*, *Lpk*, *Gck*, *Acly,* and *Fasn* measured by qPCR in the mice in (A). **(C)** Sucrose intake and plasma glucose

levels measured daily during the sucrose preference test (10% sucrose in drinking water versus water, 4 days) (*n* = 23–24 mice/group). (D) Plasma FGF21 measured after 4 days of the sucrose preference test. (E) Plasma insulin and glucose levels, and liver glycogen content in *p110α*$^{hep+/+}$ and *p110α*$^{hep−/−}$ mice fasted for 24 h and refed or not for 4 h with 20% glucose in drinking water (*n* = 6 mice/group). (F) Phosphorylation of AKT, p70S6K, and GSK3β determined by western blot in the mice in (E). (G) Relative liver expression of *Vnn1*, *Cyp4a14*, and *Fgf21* mRNA measured by qPCR in the mice in (E). (H) Acylcarnitine levels measured in the blood of *p110α*$^{hep+/+}$ and *p110α*$^{hep−/−}$ mice fasted for 24 h and refed or not for 4 h with 20% glucose in drinking water (*n* = 3 mice/group). (I) Blood glucose and ketone levels measured in *p110α*$^{hep+/+}$ and *p110α*$^{hep−/−}$ mice successively fed, fasted (24 h), and refed for 5, 10, 15, or 30 min (*n* = 6 mice/group). (J) Blood glucose and ketone levels measured in plasma from *p110α*$^{hep+/+}$ and *p110α*$^{hep−/−}$ mice fed a CTRL diet or ketogenic diet (KD) (*n* = 9–11 mice/group). (K) Representative pictures of H/E staining of liver sections from the mice in (J). Scale bar, 100 μM. Data information: In (**A–D**, **H**, **J**), data are presented as mean ± SEM. $^{#}P$ ≤ 0.05, $^{##}P$ ≤ 0.01, and $^{###}P$ ≤ 0.005 for glucose (**A, B**) or sucrose effect (**C, D**), or diet (**J**) or nutritional status effect (**H**); *$P$ ≤ 0.05, **$P$ ≤ 0.01, and ***$P$ ≤ 0.005 for genotype effect. The numerical values underlying the panels for this figure can be found in S3 Data.

In the second challenge to investigate whether p110α affects liver fatty acid uptake, we fed *p110α*$^{hep+/+}$ and *p110α*$^{hep−/−}$ mice a KD for 9 days. Mice from both genotypes showed reduced glucose levels and increased ketone bodies (Fig 3J). In addition, histological staining revealed similar tissue structures in *p110α*$^{hep+/+}$ and *p110α*$^{hep−/−}$ mice, further indicating that the absence of *p110α* does not alter dietary fatty acid uptake and catabolism (Fig 3K).

Altogether, these data suggest that hepatocyte p110α deficiency significantly influences glucose homeostasis through mechanisms that are independent of the regulation of gene expression by the carbohydrate-responsive pathways mediated by ChREBP. Moreover, p110α deficiency does not alter fatty acid uptake and subsequent regulation of liver gene expression through nuclear receptors involved in fatty acid sensing, such as PPARα.

### Role of hepatocyte p110α in the regulation of insulin-sensitive liver gene expression

Hepatocyte p110α is activated *via* multiple growth factors and their receptors. To evaluate the effect of hepatocyte p110α outside of its metabolic effect mediated by the upstream IR, we used an untargeted approach allowing combined analysis of gene expression in *p110α*$^{hep−/−}$ mice and mice lacking IR in hepatocytes (LIRKO mice) [20]. We applied a distribution transformation method to integrate datasets from the two gene expression analyses and found that the absence of hepatocyte IR is more influential on the liver transcriptome than the absence of p110α, as revealed by the amplitude of the response in LIRKO mice (up to 20) compared to the *p110α*$^{hep−/−}$ mice (up to 5) (Fig 4A). This result suggests that, in hepatocytes, p110α is not required for some of the genomic regulation downstream of IR. Consistent with previous data and in line with the role of the IR/p110α axis in the metabolic regulation of lipid biosynthesis, we found that numerous lipogenic genes (*Scd1*, *Fasn*, *Srebp1-c*, *Acly*, *Me1,* and *Gck*) are among the most commonly down-regulated genes in the absence of IR and p110α (Fig 4A). GSEA was used to identify the pathways associated with differentially expressed genes. In addition to lipid metabolism, which appears as the most down-regulated pathway in both genotypes, GSEA revealed two main clusters of pathways that are specific for p110α (green) and IR (blue) (Fig 4B). In the absence of IR, but not p110α, mice showed up-regulation of inflammatory processes, suggesting that IR regulates the inflammatory response mainly through p110α-independent signaling. In the absence of hepatocyte p110α signaling, but not IR, mice exhibited altered regulation of transcription (Fig 4B).

To investigate the regulation of hepatic gene expression that is dependent on IR, p110α, or both, we gathered regulated genes in an integrative metabolic representation highlighting insulin- and p110α-mediated transcription factor and target genes (Fig 4C). As expected, the majority of SREBP1c and FOXO1 downstream target genes are regulated through an IR/p110α-dependent axis, though the absence of IR is more influential. We also highlighted specific transcription factor/target interactions that depend almost exclusively on IR and not p110α. These pathways are related to cell proliferation (ERK, NFkB, FOXKs), lipid metabolism (HNF1α, FOXKs, Sp1), glucose metabolism (CREB, HNF6, HNF4α), and cholesterol metabolism (FXR, SREBP2). Interestingly, we highlighted CREBH as a transcription factor regulated by p110α but that is independent of hepatocyte IR. Therefore, at the level of gene expression, hepatocyte p110α acts through both IR-dependent and IR-independent pathways to regulate metabolism through the control of gene expression.

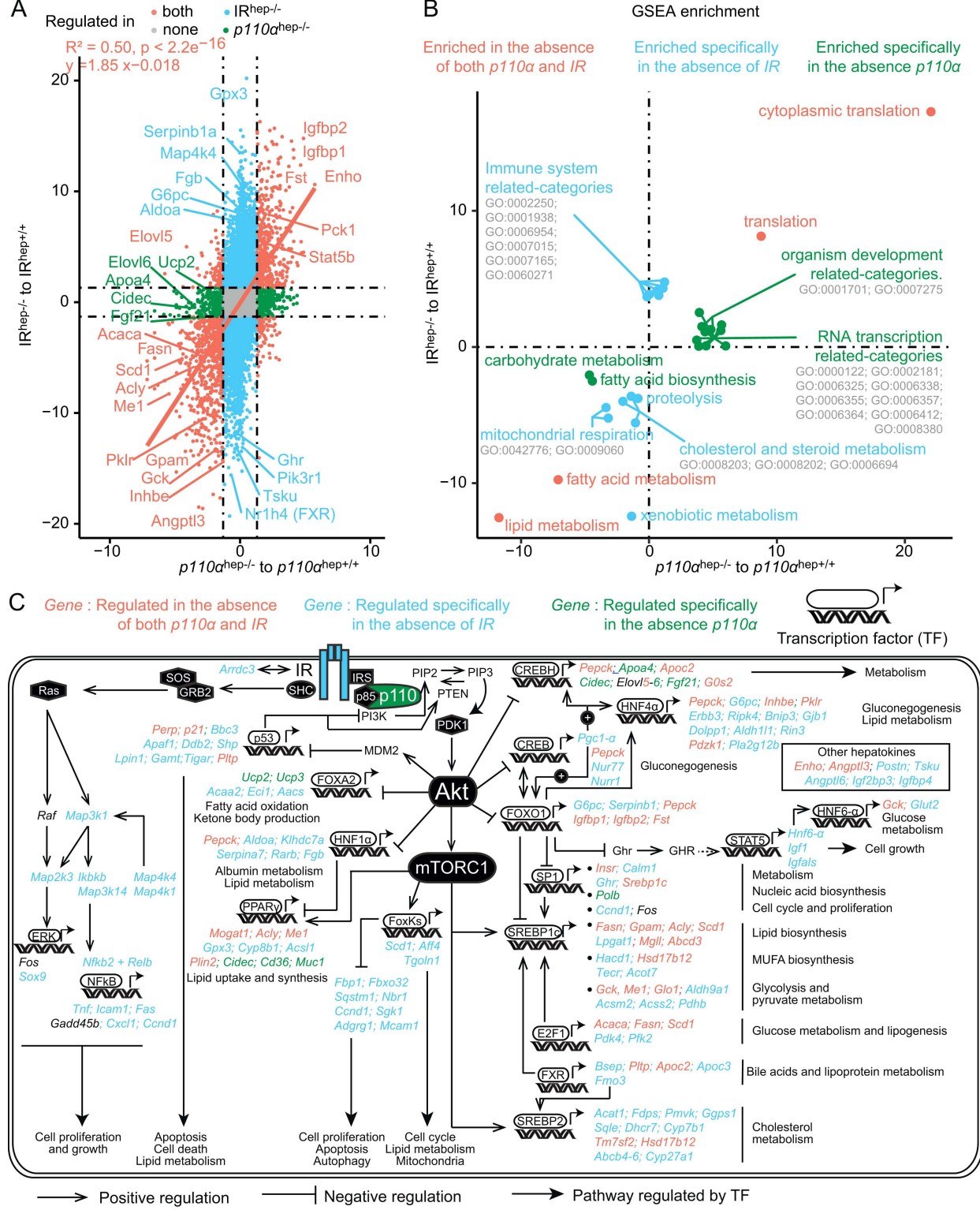

**Fig 4. Hepatocyte p110α-mediated regulation of gene expression related to IR-dependent and IR-independent pathways.** (A) Correlation plot of data extracted from two distinct microarray experiments. The values used for correlation are $-\log_{10}$ adjusted *P*-value of the comparison indicated on the axis, weighted by the sign of the corresponding $\log_2$FC of expression for each gene. (B) Correlation plot of gene set enrichment over genes presented

in (A). The values used to plot the significant categories are the –log$_{10}$ adjusted *P*-value of the comparison indicated on the axis, weighted by the sign of the corresponding normalized enrichment score. A manual annotation summarizes related GO categories when they cluster in the same area of the plot. (C) Schematic overview of insulin-responsive transcription factors and downstream-regulated genes dependent on IR, p110α, or both. Genes in blue are regulated in absence of IR but not in absence of p110α. Genes in green are regulated in absence of p110α but not IR. Genes in orange are regulated by both IR and p110α.

### Hepatocyte p110α deletion leads to insulin resistance dissociated from hepatic steatosis

Next, we evaluated the importance of p110α in hepatic changes that occur in the pathological context of obesity and MASLD, conditions in which the action of liver insulin is altered. For this purpose, *p110α^hep+/+* and *p110α^hep−/−* mice were fed a chow diet (CTRL) or HFD for 12 weeks. We confirmed that *p110α^hep+/+* and *p110α^hep−/−* mice fed a HFD develop marked glucose intolerance and insulin resistance (**Fig 5A**,**5B**). After insulin injection, the glycemia in *p110α^hep−/−* mice remained higher over the testing period than the one of *p110α^hep+/+* mice fed a HFD or of *p110α^hep−/−* mice fed a CTRL diet (**Fig 5B**). Consistently, the fasting hyperinsulinemia observed in *p110α^hep−/−* mice worsened with a HFD (**Fig 5C**).

Histology in HFD-*p110α^hep+/+* mice confirmed the accumulation of lipid droplets in hepatocytes. In contrast, *p110α^hep−/−* mice are protected from HFD-induced steatosis (**Fig 5D**). In line with histological analysis, *p110α^hep−/−* mice fed a HFD had a reduced liver weight and triglyceride accumulation compared to *p110α^hep+/+* mice, without any change in the total body weight (**Figs 5E**,**5F** and **S5**). This phenotype was associated with reduced plasma and hepatic cholesterol levels (**S5**–**S6 Fig**) and no increase in plasma ALT, a marker of liver damage (**Fig 5G**).

Dietary choline deficiency rapidly induces hepatic steatosis without obesity. To investigate the role of dietary choline in the protective effect of p110α against liver steatosis, mice were fed a CD-HFD for 12 weeks. Interestingly, despite a reduced liver weight, *p110α^hep−/−* mice had no improvement in hepatic triglycerides, inflammation, ALT and AST levels upon CD-HFD feeding (**S7**A–D Fig). Therefore, the absence of liver p110α protected mice from hepatic steatosis only with HFD feeding, not when mice exposed to a choline-deficient diet that induces steatosis without obesity. This reveals that the involvement of hepatocyte p110α in liver lipid accumulation is context-specific and is likely related to obesity and type 2 diabetes induced by HFD feeding. We also investigated the hepatic lipidome profile in the absence of *p110α* upon HFD feeding. In hierarchical clustering (**S8A Fig**), the heatmap highlighted a predominant effect of HFD that is independent from the genotype (clusters 3 and 4), but also revealed lipidomic changes with a HFD that depend on p110α (clusters 1, 5, and 6) (**S8A Fig**). The abundance of phosphatidyl ethanolamine 40:3 (PE40:3) and phosphatidyl choline 36:3 (PC36:3) was significantly reduced only in *p110α^hep−/−* mice fed an HFD, suggesting that hepatocyte p110α affects phospholipid metabolism in this context (**S8B Fig**). The abundance of PC32:1, TG51, TG55, and Cer18:0 was significantly down-regulated or up-regulated in a similar manner in *p110α^hep+/+* and *p110α^hep−/−* mice fed an HFD, suggesting a minor contribution of p110α to the regulation of these lipids (**S8C** and **S8D Fig**). PE38:4 and Cer16:0 accumulated as lipotoxic lipids only in *p110α^hep+/+* mice upon HFD feeding, and the absence of such an accumulation in *p110α^hep−/−* mice suggests protection from hepatic steatosis in these mice (**S8E Fig**). In contrast, cholesterol ester C18 (EC18) and PE36:1 species were more abundant in the absence of *p110α* upon HFD feeding and may be actively involved in the protection from hepatic steatosis (**S8F Fig**). Finally, some lipids, such as C20:5n-3 and PI36:1, were increased in *p110α^hep−/−* mice fed a CTRL diet, but the HFD erased this effect (**S8G Fig**). To investigate whether this lipid remodeling is associated with changes in liver mRNA expression, we measured the expression of genes encoding enzymes directly involved in lipid metabolism (**S8H Fig**). This analysis revealed that HFD feeding further highlights the effects of hepatocyte *p110α* deletion on lipid metabolism. In particular, cholesterol metabolism, which was almost unaffected in the absence of *p110α* upon being fed a CTRL diet (see **Fig 4**), representing an important pathway regulated by p110α in the context of a HFD (**S8H Fig**). Thus, this analysis revealed that hepatocyte p110α-dependent signaling is required for lipid remodeling that occurs during diet-induced obesity.

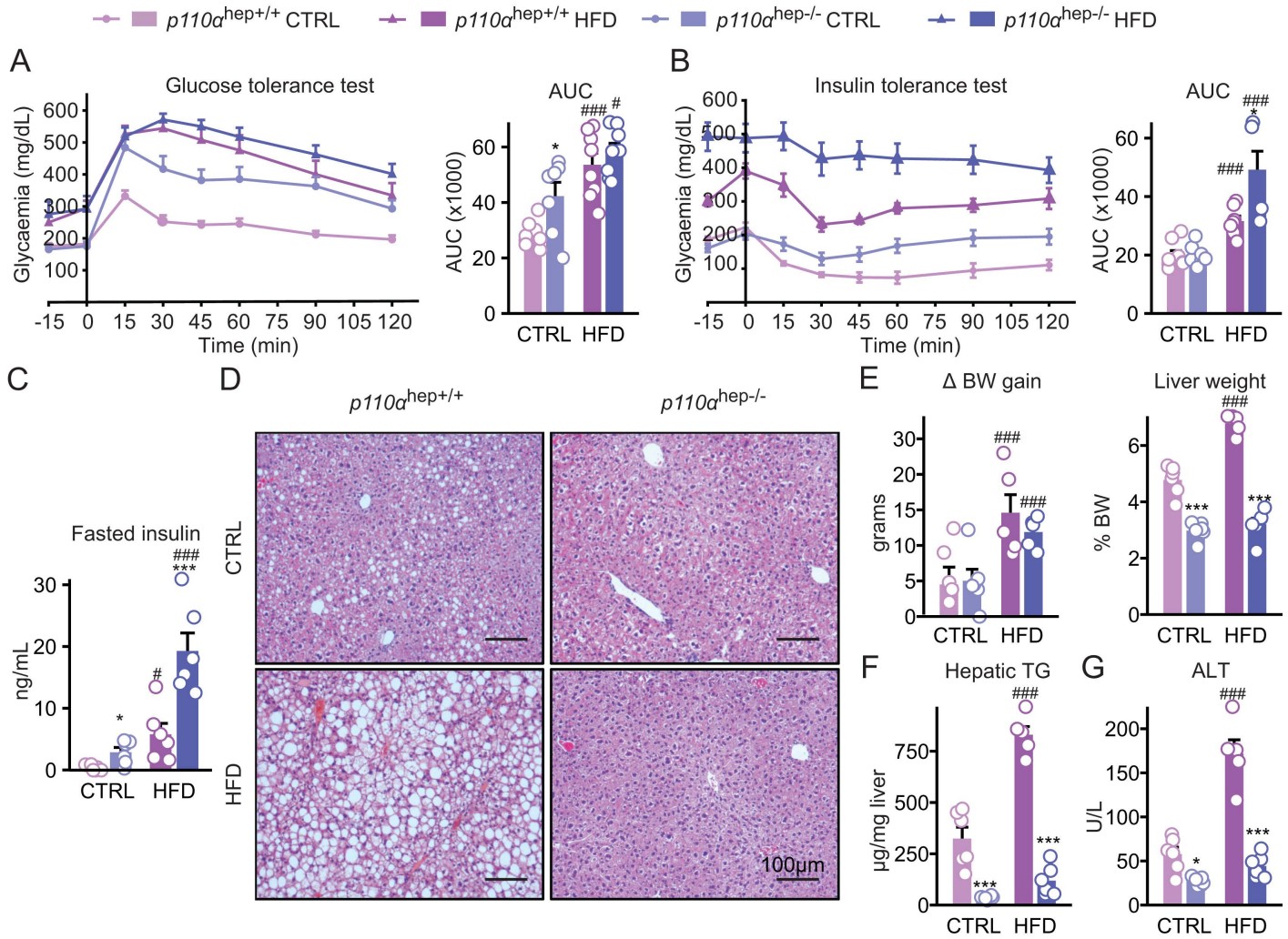

**Fig 5. Hepatocyte *p110α* deficiency disconnects hepatic steatosis from diabetes. (A,** B) Glucose **(A)** and insulin **(B)** tolerance tests and corresponding AUC for 12-week-old *p110α^hep+/+^* and *p110α^hep−/−^* mice fed a chow diet (CTRL) or HFD for 12 weeks (*n* = 6 mice/group). (C) Plasma insulin levels in 24 h-fasted *p110α^hep+/+^* and *p110α^hep−/−^* mice fed CTRL or HFD for 12 weeks. (D) Representative pictures of H/E staining of liver sections. Scale bar, 100 μm. (E) Body weight gain and liver weight at the end of the experiment in mice from (A). (F) Liver content in triglycerides. (G) Plasma ALT levels. Data information: In all graphs, data are presented as mean ± SEM. #*P* ≤ 0.05 and ###*P* ≤ 0.005 for diet effect; *P* ≤ 0.05 and ***P* ≤ 0.005 for genotype effect. The numerical values underlying the panels for this figure can be found in S4 Data.

## High fat diet reveals the critical role of p110α in the regulation of hepatic gene expression

We performed genome expression analysis to provide an overview of the role of hepatocyte p110α in the context of HFD feeding. First, we demonstrated that the vast majority of changes occurring during HFD-induced obesity are dependent on hepatocyte p110α-dependent signaling (**Fig 6A**). Genome expression analysis revealed that 3,054 (2,167 + 887) of the 3,907 genes dysregulated by HFD, i.e., 78%, are dependent on p110α (FC > 1.5 and *P* > 0.05, **Fig 6A**). Notably, under physiological conditions, p110α is responsible for only 46% of the transcriptomic changes (refer to **Fig 2B**), indicating that HFD revealed the key role of p110α in regulating the liver transcriptome. Next, we grouped genes for which expression was modified by HFD exposure in *p110α^hep+/+^* mice, representing more than 77% of all dysregulated genes. PCA revealed high transcriptomic distinction on dimension 2 between *p110α^hep+/+^* and *p110α^hep−/−^* mice exposed to an HFD, whereas

PLOS Biology | https://doi.org/10.1371/journal.pbio.3003112    April 14, 2025

17 / 32

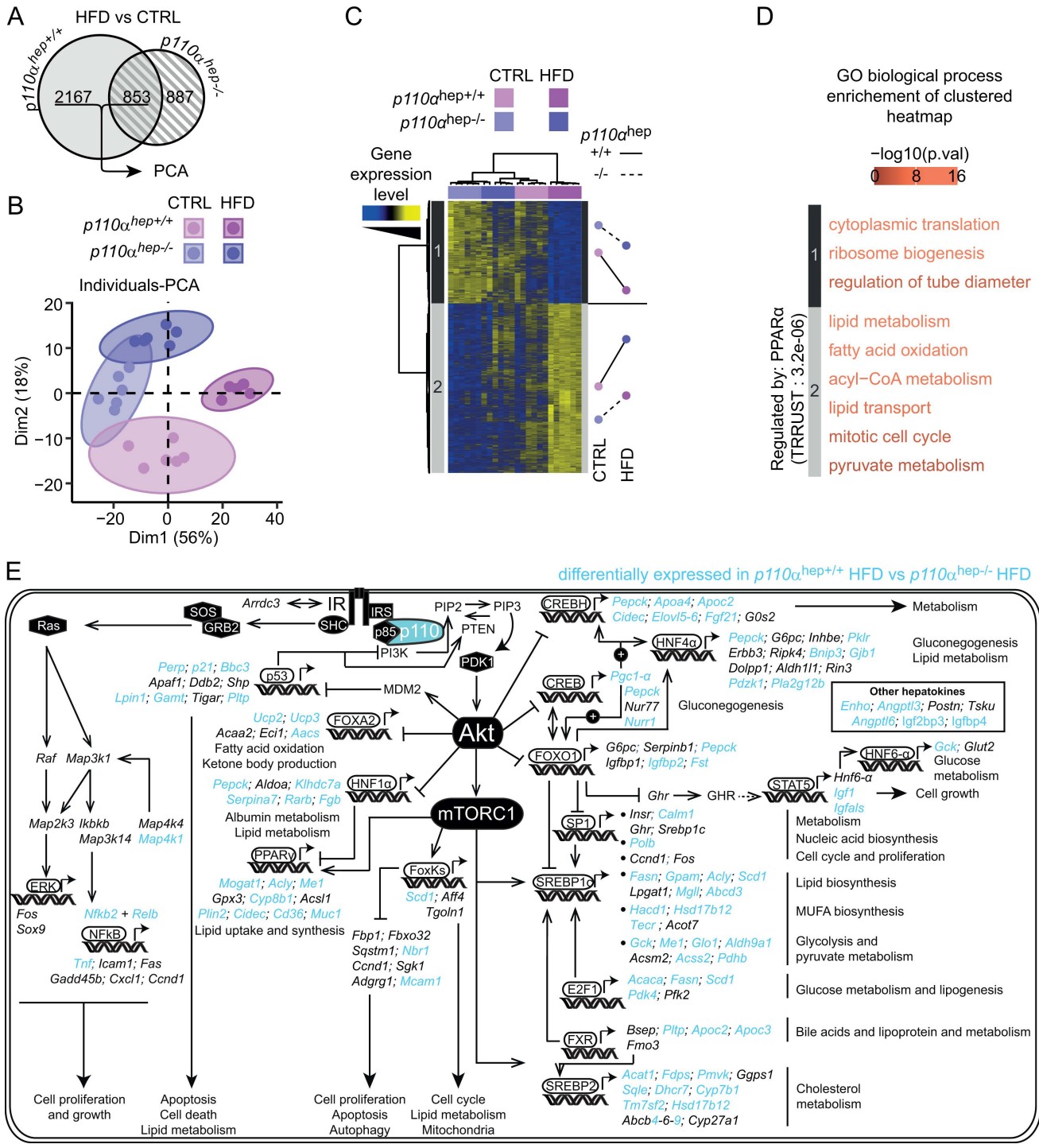

**Fig 6. Hepatocyte signaling dependent on p110α regulates HFD-mediated changes in liver gene expression profile.** (A) Euler diagram representing the number of HFD-sensitive genes determined by microarray for each genotype of 12-week-old *p110α^hep+/+* and *p110α^hep−/−* mice fed a chow diet (CTRL) or HFD for 12 weeks (*n* = 6 mice/group). (B) Principal component analysis (PCA) of genes selected in **(A)** that were regulated by HFD in both genotypes. (C) Heatmap with hierarchical clustering of genes presenting a >75% correlation to dimension 1 of the PCA in **(B)** and mean *z*-score for each group within each cluster. (D) Enrichment in GO Biological Process categories on clusters determined in **(C)** and TRRUST enrichment. (E) Schematic overview of insulin-responsive transcription factors and downstream-regulated genes dependent on or independent of p110α during HFD-induced

obesity. Genes in blue are differentially expressed in *p110α^hep−/−* mice during obesity. Genes in black are similarly expressed in *p110α^hep+/+* and *p110α^hep−/−* mice fed an HFD. The full gene expression analysis for this figure can be found in S4 Data, in the sheet "Microarray Fig 6A–C".

patterns in the gene expression of mice fed a CTRL diet remained fairly closed (**Fig 6B**). The genes were then clustered into a heatmap that further highlighted the p110α-dependent signatures during HFD-induced obesity (**Fig 6C**). Genes from cluster 1, which were highly decreased in *p110α^hep+/+* mice fed an HFD, were enriched in transcription-relative functions (**Fig 6D**). In contrast, *p110α^hep+/+* mice fed an HFD exhibited significant upregulation of genes related to fatty acid metabolism and oxidation. Hepatokines are some of the proteins significantly altered during obesity [54], which prompted us to evaluate the contribution of p110α to hepatokine regulation during HFD-induced obesity. We confirmed that multiple hepatokines, including Enho, Igfbp2, Fgl1, and Fst, are regulated through the p110α-dependent axis, and this effect is further accentuated upon HFD feeding, suggesting that p110α interferes with alterations in the communication between the liver and periphery that occurs during HFD-induced obesity (S9 Fig). We also examined the effect of HFD on the expression of genes that are under the control of transcription factors regulated by insulin. The vast majority of genes were sensitive to HFD via a mechanism involving p110α, revealing that obesity boosts the capacity of p110α to relay information from liver IR for the regulation of gene expression (**Fig 6E**). Though the IR/Akt/mTORC1 axis seemed to be further altered in the absence of p110α upon HFD feeding, the RAS/MAPK pathway remained intact in the presence or absence of p110α during HFD feeding, meaning that this pathway is largely mediated by IR-dependent but p110α-independent signaling, even during obesity. Overall, these data point to the key role of p110α signaling in hepatic transcriptomic changes that occur in HFD-induced obesity.

## Hepatocyte p110α-dependent signaling is critical for HFD-induced rewiring of gene expression

Obesity and diabetes affect the liver clock [26,31]. Given the major role of p110α in the control of HFD-induced changes in gene expression, we assessed the role of p110α in the rewiring of gene expression that occurs during obesity and MASLD. First, we confirmed that, despite unchanged body weight, *p110α^hep−/−* mice exhibited fasting hyperglycemia without an increase in liver weight and presented with similar fat depot weights upon HFD feeding (**Fig 7A**). Next, we assessed the rhythmicity of glycemia. Though *p110α^hep+/+* mice had relatively stable glycemia over the day when fed CTRL diet and HFD, with only a slight shift in the maximal peak from ZT8 (CTRL) to ZT12 (HFD), *p110α^hep−/−* mice fed a CTRL diet exhibited constant hyperglycemia over the course of the day, whereas *p110α^hep−/−* mice fed a HFD had high variation in the rhythmicity of glycemia (**Fig 7B**). By contrast, *p110α^hep+/+* and p110α^hep−/− mice had relatively consistent changes in liver glycogen content over the day when fed CTRL diet (S10 Fig). In HFD-induced obesity does is not associated with a change in liver glycogen content over the day in *p110α^hep+/+* mice but leads to a significant reduction of glycogen content in p110α^hep−/− mice.

To determine whether the absence of p110α modifies core clock genes in the context of HFD-induced obesity, we assessed gene expression of *Bmal1*, *Cry1*, *Per2*, and *Rev-erba*. The absence of hepatocyte p110α did not modify the expression of all these markers, regardless of the condition (CTRL versus HFD), suggesting that the absence of *p110α* in hepatocytes was not sufficient to alter the core clock, which remained intact in response to HFD (**Fig 7C**). Next, to further evaluate the role of p110α as a driver of HFD-induced rewiring of gene expression, we performed an untargeted analysis of liver gene expression rhythmicity in *p110α^hep+/+* and *p110α^hep−/−* mice fed a CTRL diet or HFD analyzed using the DryR package. This analysis first revealed higher circadian rhythmicity upon HFD feeding in the absence of p110α (**Fig 7D**), which is largely explained by an excessive peak of expression at the end of light (ZT10) and dark phases (ZT22) (**Fig 7E**).

We identified six distinct clusters with specific rhythmic profiles (**Fig 7F,7G**). Consistent with the aforementioned notion of an unchanged core clock, the largest cluster (cluster 1), containing 2,269 genes, regrouped genes for which rhythmicity remained intact regardless of the condition and was enriched in core clock genes. The 5 other clusters highlight the role

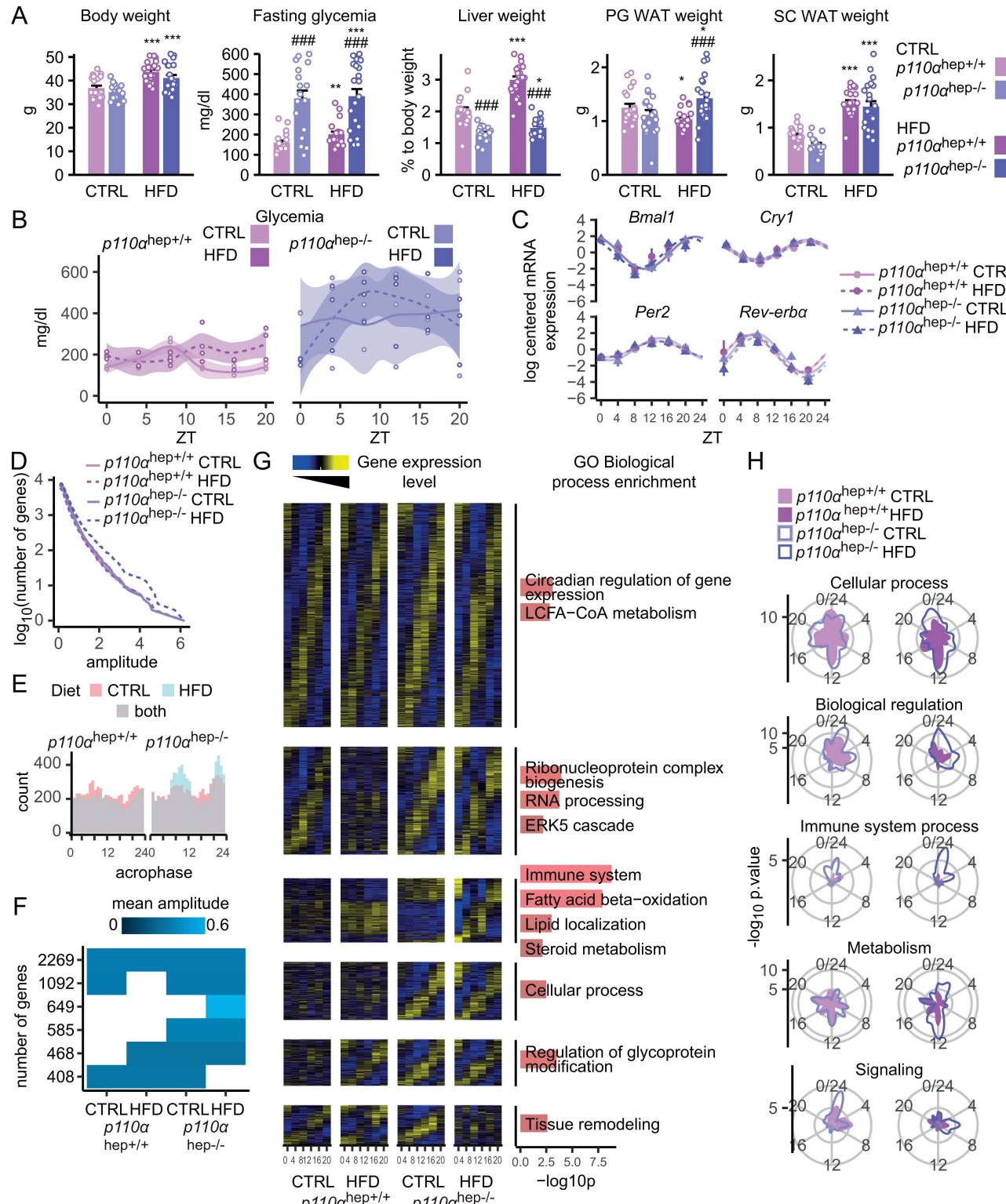

**Fig 7. p110α dependent signaling is important for the rewiring of gene expression that occurs during obesity. (A)** Body weight, fasting plasma glucose levels, and liver, perigonadal, and subcutaneous adipose tissue relative weights in 12-week-old *p110α^hep+/+^* and *p110α^hep−/−^* mice fed a chow diet (CTRL) or HFD diet for 12 weeks (*n* = 12–18/group). Data are presented as mean ± SEM. #*P* ≤ 0.05 and ###*P* ≤ 0.005 for diet effect; **P* ≤ 0.05 and

***P ≤ 0.005 for genotype effect. **(B)** Plasma glucose levels in *p110α^hep+/+* (left) and *p110α^hep−/−* (right) mice fed a CTRL or HFD diet around the clock (ZT0 to ZT20). **(C)** Relative hepatic mRNA expression levels of core clock genes and clock-controlled genes from qPCR analyzed by the Drylm function (DryR package). **(D, E)** Analysis of circadian gene expression from microarray analysis of the liver, including the cumulative number of rhythmic genes **(D)** and phase distribution of rhythmic genes **(E)**. **(F)** Representation of the six rhythmic models identified by dryR that present a significant result in hypergeo-metric testing for at least one of the first level GO Biological Process categories. Each line represents a model, and each column an experimental group. When a group has no rhythmic parameter, the corresponding tile remains empty. When two groups share the same rhythmic parameters, their tiles are colored proportionally to the mean amplitude. **(G)** Heatmap of the models selected in **(F)**. The heatmap rows, one per gene, are sorted by acro-phase. Data are scaled by row. The heatmap columns are sorted by hour (ZT0, 4, 8, 12, 16, 20). The right panel represents the GO biological process enrichment for each transcriptomic rhythmic profile. **(H)** Functional enrichment around the clock. Enrichment scores for the indicated functional terms are represented by the radial coordinate at the indicated time point. *p110α^hep+/+* mice are represented on the left and *p110α^hep−/−* mice on the right. The numerical values underlying the panels for this figure can be found in S5 Data. The full gene expression analysis for this figure can be found in S5 Data, in the sheet "DryR parameters".

of PI3Kα-dependent signaling in liver gene expression rhythmicity. A second cluster (1,092 genes) relates to genes related to RNA processing whose rhythmicity is lost in HFD when PI3Kα-dependent signaling is intact. Cluster 3 (649 genes) con-tains genes related to immunity and lipid metabolism whose hepatic rhythmicity only appears in mice lacking p110α and HFD-fed. Cluster 4 (585 genes) relates to cellular processes that become rhythmic as a consequence of p110α deficiency. Cluster 5 (468 genes) is enriched in genes related to glycoprotein modification and show rhythmicity induced by HFD in the absence of p110α. Finally, cluster 6 (408 genes) is enriched in genes related to tissue remodeling and their rhythmicity upon HFD in the absence of p110α. Interestingly, genes that became rhythmic during HFD-induced obesity in the absence of *p110α* largely belong to immune system and fatty acid beta-oxidation, and variations in the rhythmicity of these path-ways seemed to mirror feeding rhythms (**Fig 7H**). Altogether, this analysis revealed that intact hepatocyte p110α-dependent signaling impacts liver gene expression rhythmicity and its rewiring in the context of obesity and MASLD.

## Discussion

The regulation of gene expression is an essential process in controlling cell growth, survival, and metabolism in response to growth factors and hormones, such as insulin. In the present study, we investigated the contribution of hepatocyte signaling dependent on the PI3Kα catalytic subunit p110α to the control of liver gene expression in health and MASLD. Using mice with hepatocyte-specific ablation of p110α, we found that p110α is involved insulin-mediated PIP3 production *in vivo*.

Although some studies have shown that PI3Kα is entirely responsible for PIP3 production downstream of insulin [55,56], our study, along with others [15,16], suggests that hepatocyte-specific deletion of p110α leads to a significant but partial decrease in liver PIP3 production in response to insulin. This suggests that p110α is not the only mediator of insulin-dependent PIP3 production in hepatocytes [15,16,17]. Our study establishes that reduced PI3Kα-mediated PIP3 production has a profound impact on liver growth and function. In agreement with other studies [15,16,17], we found that reduced liver PIP3 is associated with a significant decrease in PDK1-dependent phosphorylation of AKT at Thr308 and mTORC2-dependent phosphorylation of AKT at Ser473. From Akt, three key pathways controlling hepatic metabolism branch out, depending on GSK3, mTORC1, and FoxO1. Insulin promotes hepatic glycogen synthesis through a mech-anism that depends on Akt-mediated inhibition of GSK3. Our analysis of hepatic GSK3 phosphorylation and glycogen content suggests that p110α is dispensable for the non-genomic control of glycogenesis by insulin and refeeding. Akt phosphorylation also activates mTORC1, another major kinase involved in cell signaling through multiple substrates, such as S6K. The AKT-mTORC1 pathway also regulates the lipogenic gene program and controls cell growth. Our data reveal that p110α deletion in hepatocytes is sufficient to impact AKT and S6K phosphorylation upon insulin stimulation, regulate gene expression related to fatty acid synthesis in the fed state, and modify relative liver weight. The sum of these data suggests that p110α deletion influences the PI3K-AKT-mTORC1 axis. Akt also inhibits FoxO1, resulting in the suppression of gluconeogenesis by downregulating the expression of proteins encoding rate-limiting enzymes in gluconeogenesis.

Based on glucose and pyruvate tolerance tests, as well as gene expression analysis, our data suggest that p110α deletion also influences the PI3K-AKT-FoxO1-dependent control of liver gene expression.

Loss of insulin signaling due to the selective deletion of the insulin receptor in hepatocytes leads to glucose intolerance associated with severe insulin resistance and strong hyperinsulinemia [57]. In young mice with a partial defect in liver insulin signaling induced by p110α deletion, we observed consistent glucose intolerance and an enhanced neoglucogenic rate. However, this was associated with only a modest reduction in whole-body insulin sensitivity. This is likely due to the limited effect of p110α deletion on fasting insulin levels and extra-hepatic insulin resistance.

Insulin signals *via* its membrane receptor to regulate gene expression through PI3K-dependent and -independent pathways [5,22]. To assess the contribution of p110α to the global gene regulation mediated by insulin, we took advantage of two existing gene expression databases that evaluate insulin-sensitive gene responses [22] and those mediated by its receptor in mice lacking insulin receptor in hepatocytes (LIRKO) [20]. We used these two independent gene expression analyses to define insulin-sensitive genes dependent on IR signaling in hepatocytes. We highlighted the contribution of hepatocyte p110α to the regulation of hepatocyte gene expression dependent on direct activation of IR signaling. These genes primarily fall into the lipid biosynthetic pathway, establishing the specific role of p110α in the IR/Akt/mTORC1 axis for the transcriptional regulation of lipid metabolism by insulin.

We further explored the influence of p110α signaling on regulators of fatty acid biosynthesis. SREBP1-c [58,59] and ChREBP [60] control the transcription of genes important for adaptations to feeding, such as those encoding for rate-limiting enzymes of *de novo* lipogenesis [21]. We found that IR/p110α-mediated lipid biosynthesis is largely dependent on SREBP1-c, but not ChREBP, as ChREBP target genes and ChREBP itself are fully induced in the absence of p110α during high glucose feeding. In contrast, SREBP1-c targets failed to increase in the absence of p110α. However, we cannot exclude that other downstream lipogenic regulators activated upon mTORC1 activation could also contribute to the regulation of *de novo* lipogenesis. One candidate is FOXK1, which is dephosphorylated, and thereby activated, following mTORC1 activation in hepatocytes [61]. Mice with hepatocyte-specific deletion of *Foxk1* exhibit reduced lipogenic activities and increased catabolic activities, which are associated with protection from hepatic steatosis, inflammation, and fibrosis in pre-clinical MASLD models [62].

A substantial portion of the insulin-sensitive and IR-dependent genes remain unchanged in the presence or absence of p110α. This could be due to liver insulin actions on gene expression that rely on other class IA PI3Ks such as p110β [17]. In addition, IR influences gene expression through pathways that do not require PI3K signaling. For example, the RAS-MAPK pathway is influenced by IR in a signaling branch that bypass the classical PI3K/Akt pathway [63]. Moreover, IR was recently described as a direct regulator of gene transcription by translocating into the nucleus [6].

In the IR/p110α comparative analysis, we highlighted some p110α-specific effects that are not related to insulin-sensitive pathways. Class IA PI3Ks integrate signals from growth factors, cytokines, and hormones other than insulin, which could explain why the absence of p110α leads to such responses. Among the panel of transcripts exhibiting p110α-dependent regulation, we highlighted a gene network related to cyclic adenosine 3′,5′-monophosphate– responsive element-binding protein, hepatic-specific (CREBH). CREBH is a membrane protein that is tethered to the endoplasmic reticulum (ER) that acts as a stress-sensing transcriptional regulator of energy homeostasis associated with hyperlipidemia, MASLD, and atherosclerosis [64]. Upon energy demands or stress, CREBH is processed to release an amino-terminal fragment that functions as a transcription factor for hepatic genes encoding lipid and glucose metabolic factors while the carboxyl-terminal fragment of CREBH (CREBH-C) is secreted as a hepatokine [65].

Overall, we have demonstrated the importance of p110α in liver growth, glucose homeostasis, and the regulation of liver gene expression including part of the insulin-sensitive regulations. In addition, we further evaluated the effects of p110α deficiency in the pathological context of obesity and type 2 diabetes obtained by feeding mice an HFD. Consistent with previous work from others, we confirmed that an absence of p110α protects mice from hepatic lipid accumulation while promoting severe glucose intolerance and insulin resistance [16]. However, studies performed in primary

hepatocytes suggested that the absence of p110α leads to defective FFA uptake [16]. Both fasting [66] and ketogenic diet feeding [67] induce significant changes in liver gene expression through processes involving PPARα that determine fatty acid catabolism, ketone body and FGF21 production [68,69]. During fasting, hepatocytes adapt to fatty acids derived from adipose tissue lipolysis [53] while in ketogenic diet feeding fatty acids originate from the diet [70]. Our work shows that in the absence of hepatocyte p110α, fatty acid catabolism and circulating ketone bodies as well as FGF21 levels are not altered during fasting and in response to ketogenic diet. This implies that liver fatty acid uptake is not affected in the absence of p110α-dependent signaling.

Three sources of fatty acids accumulate in MASLD: free fatty acids released from adipose tissue, fatty acids from *de novo* lipogenesis and dietary fatty acids [71]. Our study highlights that *de novo* fatty acid synthesis is reduced in the absence of hepatocyte p110α while fatty acid uptake and catabolism from adipose tissue or from the diet is not modified. Therefore, protection from MASLD induced by HFD feeding likely results from reduced expression of lipogenic genes and reduced *de novo* synthesis dependent on SREBP1c [72,73].

Insulin regulates the PPARα activity and ketogenesis through a mechanism involving mTORC1. Mechanistically, inhibition of mTORC1 during fasting is sufficient to induce PPARα activity [74]. Moreover, the PI3K/Akt-mTORC1/S6K axis reduces the activity of NCoR1, a transcriptional co-repressor of PPARα [75,76]. The present study demonstrates that p110α deficiency did not affect PPARα activity in feeding conditions or the capacity of refeeding to inhibit PPARα activity, despite altered phosphorylation of components downstream of insulin. These results indicate that inhibition of the p110α/Akt/mTORC1/S6K axis is not sufficient to induce PPARα activity. Thus, these findings support the notion that autonomous regulation of PPARα activity through the insulin/PI3K/Akt/mTORC1/S6K pathway is minimal compared to the induction via adipose tissue lipolysis during fasting periods [53,68,77].

Based on gene expression and lipidomic profiling, p110α is much more influential in a pathological context of HFD than under control conditions. The lipid metabolic pathway remains strongly affected upon HFD feeding in a p110α-dependent manner, which is associated with significant lipid remodeling, as revealed by the depletion of certain lipid species that massively accumulate only in fatty livers of wild-type mice. In contrast, our analysis showed an enrichment of y-linolenic acid (C18:3n-6), arachidonic acid (C20:4n-6), and eicosapentaenoate acid (C20:5n-3) in the livers of mice lacking p110α and fed an HFD when compared to wild-type animals. Interestingly, these lipid species are found at lower levels in patients with MASH compared to healthy individuals and could explain, at least in part, the phenotype of mice lacking p110α [78]. Moreover, there are numerous diet-induced models of MASLD [79,80]. We tested the effect of a choline deficient diet that efficiently induces steatosis without weigh gain. Interestingly, we found that deletion of hepatocyte p110α does not protect against steatosis in the context of CD-HFD feeding, when liver fat accumulation occurs independently from obesity. This suggests that obesity and associated insulin resistance contribute to inducing the p110α-dependent signal that promotes hepatic lipid accumulation.

Hepatocyte IR signaling is required for maintaining the rhythmicity of gene expression [20]. Therefore, we investigated whether p110α is involved in the regulation of the liver clock. First, we found that unlike genetic ablation of liver IR signaling, the deletion of hepatocyte p110α does not alter liver core clock gene expression under healthy conditions or in obesity. HFD-induced obesity was recently reported to remodel diurnal gene expression rhythmicity despite modest changes in the expression of core clock components suggesting that other clock-independent factors contribute to rhythmic liver gene expression [25,26,31]. The recent accumulation of data establishing a link between insulin signaling components, metabolic dysfunction, and the regulation of gene rhythmicity prompted us to investigate whether hepatocyte p110α contributes to the remodeling of gene expression during HFD-induced obesity [25,26,81,82]. We show that p110α-dependent hepatocyte signaling regulates rhythmic gene expression in the liver and is critical for the rewiring of gene expression that occurs during HFD-induced obesity. The p110α-dependent changes in rhythmic liver gene expression are independent of changes in the rhythmicity of liver core clock gene expression. Our analysis identifies p110α-dependent pathways as regulators of liver rhythmicity independent of the liver clock, in health and HFD-induced MASLD.

Even though our data are insufficient to make a direct link between hepatocyte p110α and downstream SREBP1-c in the regulation of HFD-induced liver gene rhythmicity, we have highlighted the critical role of insulin-driven p110α signaling in the regulation of SREBP1-c, especially during HFD-induced obesity. Thus, SREBP1c-mediated alterations in lipogenic rhythmicity during HFD feeding depends on upstream signals from the IR/p110α signaling pathway. Consistent with this idea, we show that the rhythmicity of lipid metabolism is largely modified in the absence of p110α. The modification of *de novo* lipogenesis rhythmicity during HFD-induced obesity is linked to alterations in SREBP1-c oscillations [26] and clock communications within the liver [83]. This suggests that change in PI3Kα-dependent signaling and downstream SREBP1c activity is a key regulator of obesity-induced changes in liver gene expression rhythmicity that occur in MASLD.

In conclusion, the present underlines the role of p110α in regulating key liver functions and demonstrates that hepatocyte p110α acts downstream of liver IR at multiple levels. First, hepatocyte p110α controls a part of hepatocyte PIP3 synthesis, which is sufficient to impact insulin signaling and gene expression downstream of PI3K-AKT signaling. Second, hepatocyte p110α downstream of IR and possibly other growth factor receptors is a key regulator of changes in gene expression that occur in HFD-induced MASLD, including the rewiring of gene expression rhythmicity that occurs independently of the clock. One limitation of our study is that we used a mouse model with conditional deletion of p110α, the catalytic subunit of PI3Kα. This may have led to compensation by other PI3K isotypes such as p110β, which is also expressed in hepatocytes, and may have led us to underestimate the role of p110α in hepatocytes. Further studies are needed to assess the specific roles of PI3K isotypes in liver health and in metabolic diseases.

## Supporting information

**S1 Fig. Schematic representation of the strategy for disrupting the p110α allele specifically in hepatocytes.** (A) Organization of the p110*α* targeted locus. **(B)** *p110α* targeting vector including the *loxP* sites surrounding exons 18 and 19 (length = 2 kb) and the PGK/neomycin selection cassette flanked by FRT sites and inserted between exon 19 and the second *loxP* site. **(C)** Targeted *p110α* allele containing *loxP* sites, neomycin resistance selection cassette, and FRT sites. **(D)** Targeted *p110α* allele following FLP recombinase-mediated deletion of the neomycin resistance selection cassette. **(E)** Targeted *p110α* allele following CRE-mediated deletion of floxed exons 18 and 19. Intron sequences are represented by a black line. Exon sequences are represented by filled black rectangles. The loxP and FRT sequences are represented by blue and pink triangles, respectively. The position of the primers used to validate *p110α* floxed and deleted alleles by PCR are represented by green arrows. The length of the different targeted DNA fragments is written below the schematic construction.
(PDF)

**S2 Fig. Phosphatidyl inositol profile.** (A) Phosphatidylinositol phosphate (PIP), and phosphatidylinositol biphosphate (PIP2) and phosphatidylinositol triphosphate (PIP3) relative abundance in liver samples from p110*α*hep+/+ and p110α[hep−/−] mice after fasting or under fasted conditions and treated with insulin (5 U/kg) through inferior vena cava injection (*n* = 7–8 mice/genotype/experimental condition). The analysis was performed by mass spectrometry and the relative abundance for each molecular species were calculated as a ratio to Phosphatidylinositol (PI). This method yields the number of phosphorylations and the fatty-acyl compositions of phosphatidylinositol. For example, PIP3 (C38:4) means that the phosphatidylinositol lipid has a mass that corresponds to three phosphorylations of the inositol and contains acyl chains with 38 carbons and 4 double bonds. **(B)** Quantification of signals from immunoblots, represented in Fig 1C. The numerical values underlying the panels for this figure can be found in S6 Data.
(PDF)

**S3 Fig. Transcriptomic and metabolomic analyses of p110αhep−/− mice.** (A) Enrichment analysis of the genes significantly down-regulated (left) and up-regulated (right) between p110αhep+/+ and p110 α[hep−/−] under fed conditions. **(B)** Left: Coefficient plots related to the O-PLS-DA models from [1]H-nuclear magnetic resonance (NMR) discriminating

between p110α^hep+/+ and p110α^hep−/− mice under fed conditions. The figure shows the discriminant metabolites that are higher or lower in p110α^hep+/+ versus p110α^hep−/− mice. Metabolites are color-coded according to their correlation coefficient, with red indicating a very strong positive correlation ($R^2 > 0.65$). The direction of the metabolite indicates the group with which it is positively associated, as labeled on the diagram. Right: Area under the curve of the ¹H-NMR spectra was integrated for the lactate and betaine signals. **(C)** Plasma levels of triglycerides and total, HDL, and LDL cholesterol in p110α^hep+/+ and p110α^hep−/− mice under fed or fasted conditions ($n = 6$ mice/group/experimental condition). The numerical values underlying the panels for this figure can be found in S7 Data.
(PDF)

**S4 Fig. Gene expression profile of SREBP1-c and PPARα target genes.** (A) Relative expression of Srebp1-c, Acc-α, Elovl6, Pepck, and G6pc mRNA in liver tissue from p110αhep+/+ and p110α^hep−/− mice under fasting conditions or supplemented with 20% glucose in drinking water ($n = 8$ mice/genotype/experimental condition). **(B)** Quantification of signals from immunoblots, represented in Fig 3F. **(C)** Expression of *Cyp4a10*, *Pdk4*, *Ppara*, *Fsp27*, and *Hmgcs2* mRNA in liver tissue from p110α^hep+/+ and p110α^hep−/− mice under fasting conditions or supplemented with 20% glucose in drinking water ($n = 8$ mice/genotype/experimental condition). The numerical values underlying the panels for this figure can be found in S8 Data.
(PDF)

**S5 Fig. Clinical parameters of HFD-fed p110αhep+/+ and p110αhep−/− mice under fed and fasted conditions.** Liver weight and plasma measurements including glucose levels, ketone bodies, free fatty acids, and cholesterol levels (total, HDL, and LDL). Data are presented as mean ± SEM ($n = 6$ mice per group). $^{#}P \leq 0.05$, $^{##}P \leq 0.01$, and $^{###}P \leq 0.005$ for the effect of the diet (CTRL versus HFD); $^{\$}P \leq 0.05$, $^{\$\$}P \leq 0.01$, and $^{\$\$\$}P \leq 0.005$ for the nutritional status (fed versus fasted); $^{*}P \leq 0.05$, $^{**}P \leq 0.01$, and $^{***}P \leq 0.005$ for genotype effect (+/+ versus −/−). The numerical values underlying the panels for this figure can be found in S9 Data.
(PDF)

**S6 Fig. Cholesterol metabolism in HFD-fed mice .** (A) Hepatic cholesterol and cholesterol esters levels in p110α^hep+/+ and p110α^hep−/− mice fed a CTRL diet or HFD. **(B)** Expression of Abcg5 and Abcg8 mRNA in liver tissue from p110α^hep+/+ and p110α^hep−/− mice fed a CTRL diet or HFD. The numerical values underlying the panels for this figure can be found in S10 Data.
(PDF)

**S7 Fig. Protection from hepatic steatosis in absence of p110α occurs only in the context of obesity.** (A) Body weight and plasma glucose levels in the mice from (A) in 12-week-old p110α^hep+/+ and p110α^hep−/− mice fed a chow diet (CTRL) or a choline-deficient HFD (CD-HFD) for 12 weeks ($n = 6$/genotype). **(B)** Relative liver weight and liver triglyceride content in the mice from **(A)**. **(C)** Plasma ALT and AST activity. **(D)** Representative pictures of H/E staining of liver sections. Scale bar, 100 μm. Data information: In all graphs, data are presented as mean ± SEM. $^{#}P \leq 0.05$, $^{##}P \leq 0.01$, and $^{###}P \leq 0.005$ for diet effect; $^{*}P \leq 0.05$, $^{**}P \leq 0.01$, and $^{***}P \leq 0.005$ for genotype effect. The numerical values underlying the panels for this figure can be found in S11 Data.
(PDF)

**S8 Fig. Lipidomic analysis of HFD-sensitive differences depending on p110α.** (A) Heatmap of hepatic lipid quantification in liver samples ($n = 6$/group). Hierarchical clustering highlights seven different clusters based on the lipid levels. **(B)** Relative abundance of phosphatidyl ethanolamine (PE) 40:3 and phosphatidyl choline (PC) 36:3 ($n = 6$/genotype/experimental condition). **(C)** Relative abundance of PC 32:1 and triglyceride (TG) C51 ($n = 6$/genotype/experimental condition). **(D)** Relative abundance of TG C55 and ceramide (Cer) C18:0 ($n = 6$/genotype/experimental

condition). **(E)** Relative abundance of PE 38:4 and Cer C16:0 (*n* = 6/genotype/experimental condition). **(F)** Relative abundance of esterified cholesterol C18 and PE 36:1 (*n* = 6/genotype/experimental condition). **(G)** Relative abundance of fatty acid C20:5*n*-3 and PI 36:1. **(H)** Differentially expressed genes involved in lipid trafficking, fatty acid metabolism, unsaturated fatty acid metabolism, lipid droplets, cholesterol metabolism, phospholipid metabolism, sphingolipid metabolism, and cannabinoid metabolism between *p110α^hep+/+* and *p110α^hep−/−* in mice fed CTRL diet (yellow) and HFD (grey), respectively. Data information: In all graphs, data are presented as mean ± SEM. ^#*P* ≤ 0.05 for treatment effect, *$P$ ≤ 0.05 for genotype effect. The numerical values underlying the panels for this figure can be found in S12 Data.
(PDF)

**S9 Fig. Hepatokine gene expression profile in HFD-fed mice.** The relative expression of Fgf21, Igfbp1, Gdf1, Enho, FetB, Igfbp2, Fgl1, Angptl6, Ctsd, Fst, and Lect2 mRNA in liver extracts from p110αhep+/+ and p110α^hep−/− mice fed a CTRL diet or HFD (*n* = 6 mice/group/experimental condition). The numerical values underlying the panels for this figure can be found in S13 Data.
(PDF)

**S10 Fig. Liver glycogen content of HFD-fed p110α^hep+/+ and p110α^hep−/− mice over 5 ZTs.** Quantification of liver glycogen content of HFD-fed p110αhep+/+ and p110αhep−/− mice over 5 ZTs (ZT0, ZT4, ZT12, ZT16 and ZT20). Results were normalized to the quantity of proteins for each mouse. The numerical values underlying the panels for this figure can be found in S14 Data.
(PDF)

**S1 Table. List of oligonucleotide sequences used for real-time qPCR.** List of primers used for real-time quantitative polymerase chain reaction (qPCR) analyses.
(PDF)

**S1 Data. Sheet 1, Liver PIP3 measurement in response to insulin, referenced in Fig 1B; Sheet 2, glycemic values from oral glucose tolerance test, referenced in Fig 1D; Sheet 3, glycemic values from insulin tolerance test, referenced in Fig 1E; Sheet 4, glycemic values from pyruvate tolerance test, referenced in Fig 1F.**
(XLSX)

**S2 Data. Sheet 1, Values from fasted plasma insulin, glycemia, liver weight and liver glycogen content, referenced in Fig 2A; Sheet 2, Raw data from qPCR analysis of liver samples, referenced in Fig 2F; Sheet 3, Relative abundance of fatty acids, referenced in Fig 2H.**
(XLSX)

**S3 Data. Sheet 1, Plasmatic glucose concentrations, referenced in Fig 3A; Sheet 2, Raw data from qPCR analysis of liver samples, referenced in Fig 3B; Sheet 3, Sucrose intake and glycemia over 4 days of experiment, referenced in Fig 3C; Sheet 4, Raw data from plasmatic FGF21 concentrations, referenced in Fig 3D; Sheet 5, Fasted and refed insulin, glycemia and glycogen content, referenced in Fig 3E. Sheet 6, Raw data from qPCR analysis of liver samples, referenced in Fig 3G; Sheet 7, Plasmatic acylcarnitine measurement, referenced in Fig 3H; Sheet 8, kinetic of glycemia and ketonemia, referenced in Fig 3I; Sheet 9, Glycemia and ketonemia from ketogenic experiment, referenced in Fig 3J.**
(XLSX)

**S4 Data. Sheet 1, Glycemic values from oral glucose tolerance test in CTRL and HFD fed mice, referenced in Fig 5A; Sheet 2, Glycemic values from insulin tolerance test in CTRL and HFD fed mice, referenced in Fig 5B; Sheet 3, Fasted plasmatic insulin concentrations, referenced in Fig 5C; Sheet 4, Measure of body weight gain and liver**

weight, referenced in Fig 5E; Sheet 5, Quantification of liver triglycerides, referenced in Fig 5F; Sheet 6, Measure of plasmatic transaminase ALT, referenced in Fig 5G.
(XLSX)

**S5 Data.** Sheet 1, Numerical values of body weight, glycemia, liver weight, perigonadal and subcutaneous adipose tissue weights, referenced in Fig 7A. Sheet 2, numerical values of glycemia over 6 ZTs (ZT0, 4, 8, 12, 16, 20), referenced in Fig 7c. Sheet 3, Raw data from mRNA relative abundance of Bmal1, Cry1, Per2 and Rev-erbα, referenced in Fig 7C.
(XLSX)

**S6 Data.** Sheet 1, Liver PI, PIP and PIP2 measurement in response to insulin, referenced in S2A Fig; Sheet 2, Quantification of blots illustrated in Fig 1C and referenced in S2B Fig.
(XLSX)

**S7 Data.** Sheet 1, NMR lactate and betaine measurement, referenced in S3B Fig; Sheet 2, Plasmatic concentrations of total, LDL and HDL cholesterol and triglycerides, referenced in S3C Fig.
(XLSX)

**S8 Data.** Sheet 1, Raw data from qPCR analysis of liver samples in response to glucose, referenced in S4A Fig; Sheet 2, Raw data from qPCR analysis of liver samples in response to refeeding, referenced in S4B Fig; Sheet 3, Quantification of blots illustrated in Fig 3F and referenced in S4C Fig.
(XLSX)

**S9 Data.** Sheet 1, Data of body weight, liver weight, glycemia, ketonemia, plasmatic free fatty acids, plasmatic cholesterol, HDL and LDL, referenced in S5 Fig.
(XLSX)

**S10 Data.** Sheet 1, Quantification of liver cholesterol and liver cholesterol esters, referenced in S6A Fig; Sheet 2, Raw data from qPCR analysis of liver samples in response to HFD, referenced in S6B Fig.
(XLSX)

**S11 Data.** Sheet 1, Data of body weight and glycemia, referenced in S7A Fig; Sheet 2, Data of liver weight and quantification of liver triglycerides, referenced in S7B Fig; Sheet 3, Data of plasmatic transaminases, ALAT and ASAT, referenced in S7C Fig.
(XLSX)

**S12 Data.** Sheet 1, 2, 3, 4, 5, 6, Quantification of liver lipid species, referenced in S8B,S8C,S8D,S8E,S8F,S8G Fig, respectively.
(XLSX)

**S13 Data.** Sheet 1, Raw data from qPCR analysis of hepatokines in response to HFD, referenced in S9 Fig.
(XLSX)

**S14 Data.** Sheet 1, Raw data from liver glycogen content measurement, over 5 ZTs (0, 4, 12, 16, 20), referenced in S10 Fig.
(XLSX)

**S1 Raw Images.** Raw images from PCR analysis, referenced in Fig1A and from immunoblots, referenced in Figs 1C and 3F. The list of samples corresponding to each blot is provided on pages 2 and 9 of the document.
(PDF)

## Acknowledgments

We thank Prof. Bart Vanhaesebroeck (UCL Cancer Institute) for his support and for sharing the p110α$^{flox/flox}$ mice. We thank Dr. T. Batista and Prof. C.R. Kahn (Harvard Medical School, Boston, MA, USA) for providing us with the list of insulin-sensitive hepatic genes. We thank Dr. B. Weger (EPFL Lausane, Switzerland) for his help with the use of the dryR package. We thank all members of the EZOP staff, the GeT-Trix Genotoul facility, Metatoul-Metabohub, Anexplo and We-Met facilities for their help.

## Author contributions

**Conceptualization:** Marion Régnier, Arnaud Polizzi, Tiffany Fougeray, Anne Fougerat, Phillip Hawkins, Len Stephens, Catherine Postic, Alexandra Montagner, Nicolas Loiseau, Hervé Guillou.

**Data curation:** Arnaud Polizzi, Karen Anderson, Yannick Lippi, Yuna Blum, Sandrine Ellero-Simatos.

**Formal analysis:** Marion Régnier, Arnaud Polizzi, Tiffany Fougeray, Anne Fougerat, Karen Anderson, Yannick Lippi, Sarra Smati, Céline Lukowicz, Frédéric Lasserre, Edwin Fouche, Marine Huillet, Clémence Rives, Blandine Tramunt, Claire Naylies, Géraldine Garcia, Justine Bertrand-Michel, Cécile Canlet, Sylvie Chevolleau-Mege, Laurent Debrauwer, Christophe Heymes, Thierry Levade, Yuna Blum, Sandrine Ellero-Simatos, Phillip Hawkins, Len Stephens, Alexandra Montagner, Nicolas Loiseau, Hervé Guillou.

**Funding acquisition:** Pierre Gourdy, Catherine Postic, Alexandra Montagner, Nicolas Loiseau, Hervé Guillou.

**Investigation:** Marion Régnier, Arnaud Polizzi, Tiffany Fougeray, Anne Fougerat, Prunelle Perrier, Karen Anderson, Yannick Lippi, Sarra Smati, Céline Lukowicz, Frédéric Lasserre, Edwin Fouche, Marine Huillet, Clémence Rives, Blandine Tramunt, Claire Naylies, Géraldine Garcia, Elodie Rousseau-Bacquié, Sylvie Chevolleau-Mege, Laurent Debrauwer, Christophe Heymes, Rémy Burcelin, Thierry Levade, Pierre Gourdy, Laurence Gamet-Payrastre, Sandrine Ellero-Simatos, Phillip Hawkins, Len Stephens, Catherine Postic, Alexandra Montagner, Nicolas Loiseau, Hervé Guillou.

**Methodology:** Arnaud Polizzi, Tiffany Fougeray, Anne Fougerat, Prunelle Perrier, Karen Anderson, Yannick Lippi, Sarra Smati, Céline Lukowicz, Frédéric Lasserre, Edwin Fouche, Marine Huillet, Clémence Rives, Blandine Tramunt, Claire Naylies, Géraldine Garcia, Elodie Rousseau-Bacquié, Justine Bertrand-Michel, Cécile Canlet, Sylvie Chevolleau-Mege, Laurent Debrauwer, Christophe Heymes, Rémy Burcelin, Thierry Levade, Yuna Blum, Laurence Gamet-Payrastre, Sandrine Ellero-Simatos, Julie Guillermet-Guibert, Phillip Hawkins, Len Stephens, Alexandra Montagner, Nicolas Loiseau, Hervé Guillou.

**Project administration:** Hervé Guillou.

**Resources:** Karen Anderson, Elodie Rousseau-Bacquié, Justine Bertrand-Michel, Cécile Canlet, Laurent Debrauwer, Rémy Burcelin, Thierry Levade, Pierre Gourdy, Walter Wahli, Laurence Gamet-Payrastre, Julie Guillermet-Guibert, Phillip Hawkins, Len Stephens, Catherine Postic, Hervé Guillou.

**Software:** Arnaud Polizzi, Yuna Blum.

**Supervision:** Anne Fougerat, Thierry Levade, Pierre Gourdy, Walter Wahli, Laurence Gamet-Payrastre, Sandrine Ellero-Simatos, Julie Guillermet-Guibert, Phillip Hawkins, Len Stephens, Catherine Postic, Nicolas Loiseau, Hervé Guillou.

**Visualization:** Arnaud Polizzi.

**Writing – original draft:** Marion Régnier, Arnaud Polizzi, Hervé Guillou.

**Writing – review & editing:** Marion Régnier, Arnaud Polizzi, Walter Wahli, Catherine Postic, Alexandra Montagner, Nicolas Loiseau.

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
