## [Editor Report · Decision Letter 0]

14 Nov 2024

Dear Dr Guillou,

Thank you for submitting your manuscript entitled "p110alpha-dependent hepatocyte signaling is critical for liver gene expression and its rewiring in MASLD" for consideration as a Research Article by PLOS Biology. Please accept my sincere apologies for the delay in getting back to you with feedback.

Your manuscript has now been evaluated by the PLOS Biology editorial staff and I am writing to let you know that we would like to send your submission out for external peer review.

Once your full submission is complete, your paper will undergo a series of checks in preparation for peer review. After your manuscript has passed the checks it will be sent out for review. To provide the metadata for your submission, please Login to Editorial Manager (https://www.editorialmanager.com/pbiology) within two working days, i.e. by Nov 16 2024 11:59PM.

Kind regards,

Richard

Richard Hodge, PhD

rhodge@plos.org

PLOS

---

## [Decision Letter · Decision Letter 1]

22 Jan 2025

Dear Hervé,

Thank you for your continued patience while your manuscript "p110alpha-dependent hepatocyte signaling is critical for liver gene expression and its rewiring in MASLD" was peer-reviewed at PLOS Biology. Please accept my sincere apologies for the delays that you have experienced during the peer review process. Your manuscript has now been evaluated by the PLOS Biology editors, an Academic Editor with relevant expertise, and by two independent reviewers.

In light of the reviews, which you will find at the end of this email, we would like to invite you to revise the work to thoroughly address the reviewers' reports.

As you will see, whilst both reviewers note that the metabolic phenotypic effects of liver-specific PI3Ka deletion have been previously reported, they agree that the insights provided into the transcriptional networks regulated by insulin and PI3K are interesting. Reviewer #1 raises specific concerns around the lack of quantifications for the western blots and the effect sizes reported for insulin tolerance, whereas Reviewer #2 notes that the mouse model may be subject to functional compensation by other PI3K isoforms and asks that additional discussions are provided.

Given the extent of revision needed, we cannot make a decision about publication until we have seen the revised manuscript and your response to the reviewers' comments. Your revised manuscript is likely to be sent for further evaluation by all or a subset of the reviewers.

**IMPORTANT - SUBMITTING YOUR REVISION**

*Re-submission Checklist*

*Published Peer Review*

*PLOS Data Policy*

*Blot and Gel Data Policy*

Best regards,

Richard

Richard Hodge, PhD

rhodge@plos.org

REVIEWS:

Reviewer #1: The metaboic phenotype of liver-specific PI3Kalpha knockout have been previously described (see Ref 16 of this manuscript).

The manuscript "p110alpha-dependent hepatocyte signaling is critical for liver gene expression and its rewiring in MASLD" by Marion Régnier et al. investigates the role of PI3Kalpha within the hepatocyte in the control of liver gene expression including the reprogramming induced by HFD, fasting and reefeding, and circadian gene expression. The manuscript is overall interesting but some issue needs to be addressed before publication.

1) in the abstract the authors claim that most of PI3K activity induced by insulin is dependent on PI3Kalpha. However, in Figure 1 B the author show that loss of PI3Kalpha in hepatocyte has no effect of baseline PIP3/PIP2 and reduces by 50% the maximal insulin-induced PIP3/PIP2. Therefore it should be concluded that loss of PI3Kalpha has no effect on baseline PI3K activity but it reduces maximal insulin-stimulated PI3K activity.

2) The immunoblots in Figure 1 C with n=2 are not quantitative. The authors should repeat the blots with at least n=3/4 quantify bands intensities and perform statistical analysis.

3) The data in figure 1E indicate a minor effect of loss of PI3Kalpha in hepatocytes on insulin tolerance. how the authors reconcile this observation with the more pronounced difference in glucose and piruvate tolerance?

4) Figure 2 A should include liver glycogen content.

5) The immunoblots in Figure 3F with n=2 are not quantitative. The authors should repeat the blots with at least n=3/4 quantify bands intensities and perform statistical analysis.

6)Figure 3 should include fasted and refed liver glycogen content.

7) Figure 5 should show CTRL and HFD liver glycogen content ideally over 4 ZT time points covering the 24 hrs liver glycogen rhythm.

Reviewer #2: The manuscript by Régnier et al. reports phenotypes of liver-specific deletion of PI3Ka upon HFD and other feeding-induced metabolic challenges.

The phenotypic effects of liver-specific deletion of PI3Kα—such as the development of insulin resistance and protection from liver steatosis upon high-fat diet (HFD) feeding—have been reported in several publications, which are cited in the current manuscript. Nevertheless, the study under review has significant merit. It presents a comprehensive transcriptomic analysis of PI3K's role in regulating liver gene expression, comparing and contrasting the effects of PI3Kα deletion with those of insulin receptor deletion. These findings are further supported by an extensive integration of metabolite profiling data with the gene expression results.

Among its key findings, the study demonstrates that PI3Kα does not regulate ChREBP or PPARα but is involved in the regulation of CREBH, intriguingly in an insulin receptor-independent manner. This represents a novel and noteworthy discovery. Additionally, the study implicates PI3Kα in the regulation of rhythmic gene expression in the liver.

Overall, these data provide a meaningful contribution to the field of insulin signaling research, offering deeper insights into the role of PI3K in metabolic regulation. While the study is generally well-presented, a few areas require clarification and improvement. Also, additional proofreading is needed to enhance figure legends and labeling for greater clarity.

In terms of criticisms, a limitation of the gene-targeted mouse model used in this study, which is effectively a conditional knockout, is that such models can exhibit partial functional compensation by other isoforms when studying pathways with multiple isoforms, such as the PI3K pathway. The ability of another isoform to compensate in an artificial system where the targeted isoform (PI3Kα in this case) is physically absent (as in a knockout) does not necessarily imply that the compensating isoform could fulfill the same role physiologically in a normal cell, where the PI3Kα isoform is intact. Consequently, conditional knockout models may not accurately represent the physiological function of the targeted protein compared to catalytically inactive point mutants or treatments with isoform-selective inhibitors. Unfortunately, many studies utilizing conditional knockout PI3K models fail to acknowledge this limitation, detracting from their scientific precision.

Discussion Section: The statement, "Moreover, the present study suggests that hepatocyte p110α selectively phosphorylates stearoyl/arachidonoyl species of PIP2 to produce PIP3 in response to insulin," likely refers to the phosphoinositide profiling shown in Supplementary Figure 2. However, the legend for this figure lacks sufficient detail to clarify the designations in the graphs. Furthermore, the interpretation of these data in the corresponding sections of the main text is minimal and would benefit from further elaboration.

Minor issues:

Main text cites 'Fig 8' instead of the correct 'Fig. 7'

Fig 7A statistical significance comparisons is unclear.

Fig 7H legend: It is not clear what is meant by "rows" and "columns"

Suppl Fig. 5 lacking explanation of statistical comparisons

Suppl Fig. 6 panel C lacking a legend description

---

## [Editor Report · Decision Letter 2]

12 Mar 2025

Dear Hervé,

On behalf of my colleagues and the Academic Editor, Rebecca Haeusler, I am pleased to say that we can accept your manuscript for publication, provided you address any remaining formatting and reporting issues. These will be detailed in an email you should receive within 2-3 business days from our colleagues in the journal operations team; no action is required from you until then. Please note that we will not be able to formally accept your manuscript and schedule it for publication until you have completed any requested changes.

PRESS

Best wishes, 

Richard

Richard Hodge, PhD

rhodge@plos.org

PLOS
